

# Retention During Freezing of Raindrops, Part II: Investigation of Ambient Organics from Beijing Urban Aerosol Samples

Jackson Seymore*[1], Martanda Gautam[1], Miklós Szakáll[1], Alexander Theis[2], Thorsten Hoffmann[2], Jialiang Ma[3], Lingli Zhou[4], Alexander L. Vogel[3]

[1] Institute for Atmospheric Physics, Johannes Gutenberg University, Mainz, Germany
     [2] Particle Chemistry Department, Max Planck Institute for Chemistry, Mainz, Germany
     [3] Institute for Atmospheric and Environmental Sciences, Goethe University Frankfurt, Germany
     [4] South China Institute of Environmental Sciences, Ministry of Ecology and Environment, Guangzhou, P.R. China
     *Corresponding author: seymorej@uni-mainz.de

**Keywords**: retention, dissolved organic matter, DOM, Orbitrap MS, secondary organic aerosol, SOA, convective clouds, acoustic levitator

## Abstract

The freezing of hydrometeors incurs certain water-soluble organic compounds dissolved in the supercooled cloud droplets to be released into the gas phase. This may lead to the vertical redistribution of

substances that become available for new particle formation in the upper troposphere. Drop freezing experiments were performed on the Mainz Acoustic Levitator (M-AL) using aqueous extracts of ambient samples of Beijing urban aerosol. The retention coefficients of over 450 compounds were determined. Most nitroaromatics and organosulfates were fully retained along with the aliphatic amines (AA) and higher-order amines and amides while sulfides, lipids, aromatic hydrocarbons, and long chain compounds

are among the most unretained and incidentally the fewest species observed. The findings here also indicate that $NO_x$ and $SO_x$ chemistry, particularly anthropogenically related, enhances the retention of the resulting secondary organic aerosols (SOA). A positive correlation between polarity and freezing retention along with a negative correlation with vapor pressure and freezing retention was observed. No sigmoidal relationship with effective Henry's law constant was observed which differs with the

parameterizations of riming retention presented in current literature, which is justified by the lower surface-to-volume ratio of the large drop size investigated.



## 1 Introduction

Atmospheric organic matter (OM) plays a critical role in climate regulation directly through radiation scattering and indirectly through cloud condensation nucleation which impacts Earth's energy

balance through radiative forcing (Fofie et al., 2018; Liu et al., 2018). These effects are controlled by factors such as their optical properties, size, and the hygroscopicity (Dusek et al., 2006; Sun et al., 2021), which can change based on the proportions of primary organic aerosols (POA)—directly emitted aerosols—and secondary organic aerosols (SOA)—aerosols formed from the oxidation products of volatile organic compounds (VOC) as part of new particle formation (NPF) (Hallquist et al., 2009; Liu et

al., 2021; Riva et al., 2019). Convective systems have been suggested to support NPF in the outflow region by reducing existing particle concentrations, facilitating cold temperatures, and transporting reactive gases into regions with high actinic fluxes (Clarke et al., 1998; Zheng et al., 2021)

Nucleation-mode particles—with sizes in the lower tens of nm—have consistently been observed in concentrations of up to $10^4$ cm$^{-3}$ from aircraft in the upper troposphere (Andreae et al., 2018; Andrés

Casquero-Vera et al., 2020; Clarke et al., 1999; Heitto et al., 2024; Weigel et al., 2011; Williamson et al., 2019). These measurements significantly exceed the corresponding concentrations in the planetary boundary layer and indicate that the main source of such ultra-fine particles in the upper troposphere is in situ NPF rather than their direct transport from the boundary layer (Bardakov et al., 2021). The traditional explanation for this phenomenon has been that the reduction of existing aerosol particles in deep

convective clouds eliminates removal processes for small particles and condensable vapors, supporting NPF (Clarke et al., 1998). However, Williamson et al. (2019) also showed that even without these conditions, such as in tropical convection, these newly formed particles can still be found. They then argue that most models underestimate available organic matter at high altitudes and as a result predict less NPF in these regions. If this NPF is the result of an overlooked mechanism of organic matter transport, it

is then critical to elucidate this mechanism so to constrain uncertainty around the influence of high altitude NPF from convective outflows (Bardakov et al., 2021).



Publications by Borchers et al. (2024) and Jost et al., (2017) have demonstrated a potential

mechanism for organic matter transport in mixed-phase clouds. They describe how organic compounds

that are exchanged between the gas and aqueous phase in cloud droplets can either be trapped in the ice

phase during freezing—washing them out by precipitation—or return to the gas phase by volatilization.

This revolatilization incurred by the freezing process leads to a vertical redistribution and has the

potential to explain the occurrence of organic matter at high altitudes in regions with deep convection.

Alternatively, earlier publications by Pruppacher and Klett (2010) and Snider and Huang (1998) have

suggested that complete sublimation of ice particles can also transport 'retained' compounds trapped in

the ice phase and release them at high altitudes.

In convective clouds the main formation pathway of precipitation-size ice particles is riming, i.e.

the freezing of supercooled μm-sized cloud droplets on the surface of a mm-sized ice particle. Thus,

riming retention is an important process for the vertical redistribution of water-soluble organic

compounds (WSOC). Convective clouds with warm bases favor the formation of mm-sized drops by

collision-coalescence (Lamb and Verlinde, 2011), which subsequently can be uplifted in the updraft to

regions with temperatures below 0°C. Once beyond the freezing level they can freeze and thereby release

dissolved matter into the gas phase. This identifies freezing retention of mm-sized drops as a potential

contributor to the vertical redistribution of WSOCs and was experimentally investigated in the present

study.

The proportion of a substance that remains in the ice during this phase change is described by the

retention coefficient R, which indicates the percentage of the trapped substance with a value between 0

and 1 (Bela et al., 2018; Iribarne and Pyshnov, 1990; Snider et al., 1992; Stuart and Jacobson, 2004). A

species' retention is influenced by its chemical properties, such as its dimensionless effective Henry's law

solubility constant (H*), as well as the physical properties of the droplet such as temperature, liquid water

content, droplet size, and ventilation. Substances with a small H* are more likely to return to the gas

phase during riming, which results in a lower retention coefficient. Additionally, these external and



physical conditions of the droplet disproportionally influence the retention for these small H* substances (Jost et al., 2017; Stuart and Jacobson, 2003, 2004)

Current experimental studies to determine retention coefficients for atmospheric constituents and relevant SOA precursors have focused on inorganic species, small organics, or single component mixtures with significantly higher than natural concentrations. Additionally, current studies have only examined retention in droplets within natural cloud size range rather than raindrop sizes. The few studies that look at complex mixtures are limited to compounds of similar families and only a handful of species (Borchers et al., 2024). Naturally occurring atmospheric constituents that are observed in rainwater are present as complex mixtures of potentially thousands of species (Seymore et al., 2023). To get closer to observing the retention of compounds in their natural conditions, this study presents measurements of retention coefficients for a real, complex mixture of WSOC extracted from filter samples taken in an urban environment.

## 2 Methods

### 2.1 Sampling Location and Method

A high-volume sampler (HiVol) was run with quartz fiber TSP filters over three nights between March 3 and 5, 2022 in Beijing, China (40.0426° N, 116.4197° E) for an approximate sample volume of 550 $m^3$ between the hours of 21:00 to 9:00. These filters were sealed in aluminum foil and stored at -20°C until analysis. Aqueous extracts of these filters were prepared by taking 1/4 of each filter, combining them (in total 3/4 of a 203 x 254 mm filter area) in 30 ml Milli-Q water, and then extracting with an orbital shaker for 15 min. The same was performed for a blank sample; a total 3/4 of unsampled filter area was extracted in 30 ml Milli-Q water for 15 min. These extracts were filtered through a 0.2 µm PTFE filter. 10 ml of the prepared extract was reserved for Ultra-High Performance Liquid Chromatography High-



100 Resolution Orbitrap Mass Spectrometry (UHPLC-HRMS) analysis and stored at 3 °C while the remaining

20 ml was sent to Institute for Atmospheric Physics at the Johannes Gutenberg University of Mainz,

Germany for freezing experiments.

## 2.2 Mainz Acoustic Levitator (M-AL)

Freezing experiments were performed in an acoustic levitator (APOS BA 10, tec5 GmbH). This

allows contact-free single-drop levitation maintained by a standing ultrasonic wave. This setup and its

relevant physical influences are described in detail by Diehl et al., (2014), Szakáll et al., (2021), and in

part 1 of this publication series by Gautam et al. (2024). For the freezing experiment, the M-AL is placed

inside a walk-in cold room where the ambient temperature was set to -15°C. The M-AL is surrounded by a

protective acrylic housing to prevent any disturbance from air motion which may cause unsteady

temperature conditions, unstable levitation, or carry unwanted ice-nucleating particles onto the drop

surface. Air temperature in the M-AL was measured by a PT100 sensor and an infrared thermometer (KT

19.82 II, Heitronics) was used to monitor drop surface temperature.

The aqueous filter extract was injected with a syringe into the M-AL node to form a single free-

floating drop with a diameter of $2 \pm 0.1$ mm. The drop was allowed to freeze without the introduction of

artificial freezing nucleator. Freezing time was approximately 90 seconds on average but not longer than 3

min. Once the drop was fully frozen, it was removed from the M-AL and stored in a

polytetrafluoroethylene (PTFE) vial at –20°C until analysis. Enough drop to reach the minimum viable

sample volume for analysis, 50 µl, were collected to produce a single sample (approximately 12 drops).

Two full samples were collected for UHPLC-HRMS analysis. The blank filter extract was also used in the

freezing experiment to produce two more travel blank samples for comparative analysis and background

subtraction.



### 2.3 UHPLC-HRMS analysis

In addition to the M-AL frozen extract and the travel blanks, 100 µl of the reserved extract and

Milli-Q solvent was analyzed by UHPLC-HRMS. Chromatographic separation was performed (Vanquish

Flex, Thermo Fisher Scientific Inc.) on a reversed phase column (Cortecs Solid Core T3, 2.7 µm, 150 × 3

mm, with the corresponding VanGuard Cartridge, Waters Corp.). Samples were ionized in negative and

positive mode using a heated electrospray ionization source (HESI-II Probe, Thermo Fisher Scientific

Inc.) and then detected with a high-resolution hybrid quadrupole-Orbitrap mass spectrometer (Q Exactive

Focus, Thermo Fisher Scientific Inc.). The chromatographic settings and gradient are as follows: LC

solvent A: ultrapure water with 0.1% formic acid; LC solvent B: methanol with 0.1% formic acid; Flow

rate 400 µl min$^{-1}$; pre-column heater and post-column cooler 40 °C; Gradient: 0 min 1% B; 1 min 1% B;

15 min 99% B; 16.5 min 99% B; 17.5 min 1% B; 20 min 1% B. The MS settings were at fullMS scan

(*m/z* 50-750; resolution 70k) along with data-dependent MS2 in discovery mode (resolution at 17.5k) for

acquiring fragmentation spectra of the largest peaks.

### 2.4 Non-targeted Analysis and Property Estimation

Compound identification confidence is communicated here using the convention described in

Schymanski et al., (2014). The raw UHPLC-HRMS files were processed on Compound Discoverer 3.3

(Thermo Fisher Scientific). This software aligned chromatographic peaks of interest with a maximum

shift of 0.1 min in retention time and a mass tolerance of ±2 ppm. Mass traces with retention times less

than 1.8 min were excluded as they are not considered to be chromatographically separated. Ions were

detected if the peak intensity was at least $5 \times 10^5$ counts for [M−H]$^-$ for negative mode or [M+H]$^+$ and

[M+Na]$^+$ for positive mode. In addition to the mass-to-charge ratio of the detected ion, at least one

corresponding isotopologue had to be measured. The tolerance between the measured and calculated

intensity of the isotopologue was less than 30 %. These unknown compounds were then grouped with a



retention time tolerance of 0.1 min to produce a merged MS feature and those of them with a sample-to-blank ratio smaller than 5 were marked as background and removed from the dataset. Any compounds

that did not appear in the reserved sample of filter extract were also removed from the dataset. A peak quality score was given on a scale of 0 to 10—with 10 being a perfect chromatographic peak—for each mass trace based on its peak shape qualities, e.g. peak jaggedness, modality. For all mass traces with a peak quality higher than 6 in all samples, a predicted composition for each mass trace was calculated within ±2 ppm with the allowed elements of carbon (C), hydrogen (H), nitrogen (N), oxygen (O), and

sulfur (S). Compounds were grouped together as CHO, CHNO, CHOS, and CHNOS. It is important to note that phosphorus (P) containing species were not considered for this study. All level 5 (L5) or higher compounds including any mass traces that did not fit a predicted composition within ±2 ppm were used for calculating retention coefficients using their integrated MS signals. Level 4 (L4) or higher compounds with determined compositions were used for Van Krevelen and Kroll analysis to highlight the validity of

the dataset as a real, complex mixture of urban-influenced WSOC. To aid the visualization of MS data, a Van Krevelen diagram cross-plots the H:C ratio as a function of the O:C ratio while a Kroll diagram cross-plots the estimated average carbon oxidation state as a function of the number of carbon atoms. An estimated vapor pressure at 298 K was then calculated for the elemental composition based on the parameterization by Li et al. (2016).

165        The predicted compound list was then matched against the mzCloud database (HighChem LLC, 2013-2021) for comparing MS$^2$ spectra. If a compound had at least one positive match with the predicted compound in either database as well as a peak quality score above 8 in the reserved extract, this level 3 (L3) or higher tentative candidate was selected to be used for calculating its effective Henry's law constant ($K_H$, mol Pa$^{-1}$ m$^{-3}$). These properties were predicted using the HENRYWIN$^{TM}$ model as part of

the EPI Suite$^{TM}$ package which provides the values at 298 K (US EPA, n.d.). If the EPI Suite$^{TM}$ was able to find an experimental value based on a CAS lookup match, those values were used over the model



prediction. This is only applicable to the minority of identified species. The effective Henry's law

solubility was converted to a dimensionless effective Henry's law constant (H*) using the equation:

$$H^* = K_H * \bar{R}T \tag{1}$$

Where $\bar{R}$ is the ideal gas constant (8.3144626 $m^3$ Pa $K^{-1}$ $mol^{-1}$) and T is temperature (K). This conversion

allows for a dimensionless comparison and considers dissociation and hydration effects. This calculation,

however, requires structural information about a compound. As a result, performing this calculation on a

L3 tentative structural candidate can be specious or misleading. Regardless, H* for structural isomers

using HENRYWIN$^{TM}$ typically differs less than 3 orders of magnitude with an overall average of

approximately 1.5. This accuracy is sufficient for this analysis (Isaacman-Vanwertz and Aumont, 2021).

### 2.3 Retention Calculation and Tracer Corrections

The signals of species that also appeared in the travel blanks were first subtracted from the M-AL

samples to remove their ambient signal but remained in the dataset as they exceeded the sample-to-blank

ratio of 5 and could not be considered background. A naturally occurring tracer was selected from the

dataset for both positive and negative mode to correct for any dilution, evaporation, and desorption that

may occur. To be an ideal tracer, the compound should be fully retained during freezing and have an

adequate MS signal. For this work, the peak quality was required to be higher than 8 in the reserved

extract and for there to be a positive database match for the predicted composition. For negative mode,

this tracer was 4-Nitrophenol ($C_6H_5NO_3$, 139.0269 *m/z*, 9.7 min, L3). This was chosen as previous

experiments by Borchers et al., (2024) have identified this compound to have a retention coefficient of

1.01 ± 0.07 during riming experiments, where desorption and evaporation effects are likely to be more

influential than in the present experiment due to enhanced ventilation and much smaller droplet size. For

positive mode, xylitol ($C_5H_{12}O_5$, 152.0685 *m/z*, 1.6 min, L3) was selected as the tracer as its H* has been

determined to be higher than $10^8$ (Compernolle and Müller, 2014), which according to Borchers et al.,



(2024) and Jost et al., (2017) indicate its retention coefficient can be safely assumed to be 1, even under the higher exchange conditions of wind tunnel experiments. Regardless of the accuracy of the L3-assigned structure, these tracers represent species with retentions very close to 1 that allow for a reference to that value to compare between samples and make corrections for non-freezing mass exchange.

The equation for calculating the retention coefficient is adopted from Borchers et al., (2024) and Jost et al., (2017) and used here as

$$R = \frac{S_{\text{substance}}^{\text{sample}} / S_{\text{substance}}^{\text{RES}}}{S_{\text{tracer}}^{\text{sample}} / S_{\text{tracer}}^{\text{RES}}} \qquad (2)$$

where the numerator describes the ratio between the peak area of the substance of interest in the ice sample ($S_{\text{substance}}^{\text{sample}}$) and in the reserved extract sample ($S_{\text{substance}}^{\text{RES}}$). The denominator describes the same

ratio but for the tracer ($S_{\text{tracer}}^{\text{sample}} / S_{\text{tracer}}^{\text{RES}}$). Since no dilution effects are involved in the measurement and all samples are measured in the same aqueous matrix, signal ratios can be compared directly without a calibration curve, provided that detector response is linear within the given range of measurement. Given that ions below the threshold intensity of linearity were excluded from measurement, dynamic mass calibration of the HRMS was performed prior to measurement, and that HRMS instruments of this

generation show linear dynamic ranges of at least five orders of magnitude (Kaufmann and Walker, 2017), it is reasonable to assume linearity over the measurement range. As ventilation and evaporation effects are quite low in the M-AL (Szakáll et al., 2021), their effects are compensated for by the tracer. Compensating for desorption effects is more complex. As desorption is thought to be driven mainly by linear diffusion and thus enhanced by increased ventilation, species with higher than ambient

concentrations as well as species with higher vapor pressures are thought to be disproportionally influenced. However, since the M-AL has little ventilation to enhance desorption, HiVol filter sampling already bias against high vapor pressure species (Bidleman et al., 2020), and ambient filter sampling is closer to ambient concentrations relative to previously simulated single component mixtures, desorption



effects are only compensated for by the tracer and nonuniform desorption effects are considered

negligible.

Further data analysis was performed with MATLAB ver. R2023a. Distribution modeling was

performed using the Distribution Fitter from the Statistics and Machine Learning Toolbox 12.5, based on

the Stable Distribution and t Location Scale Distribution models provided in *Univariate Stable*

*Distributions* (2020) and *Univariate Continuous Distributions* (2015) (Nolan, 2020; Yee, 2015).

## 3    Results and Discussion

### 3.1 Dataset Description

In the negative mode, from over 2800 MS features measured, 548 significant, non-background

detected compounds were found. 208 met the peak quality constraints and were then used for analysis.

196 compounds (94%) had successfully assigned compositions and 77 were then selected for additional

230    property calculations.

In the positive mode, 342 significant, non-background detected compounds were found from over

1800 features. 250 met the peak quality constraints and were then used for analysis. 218 of those

compounds (87%) had successfully assigned compositions. 84 were then selected for additional property

calculations. Comparatively fewer compounds were assigned compositions in the positive mode as

phosphorous containing species were not considered. These can represent almost a third of positively

ionizable species in rainwater WSOC (Seymore et al., 2023) so it likely makes up a significant portion of

species variety that is not considered.

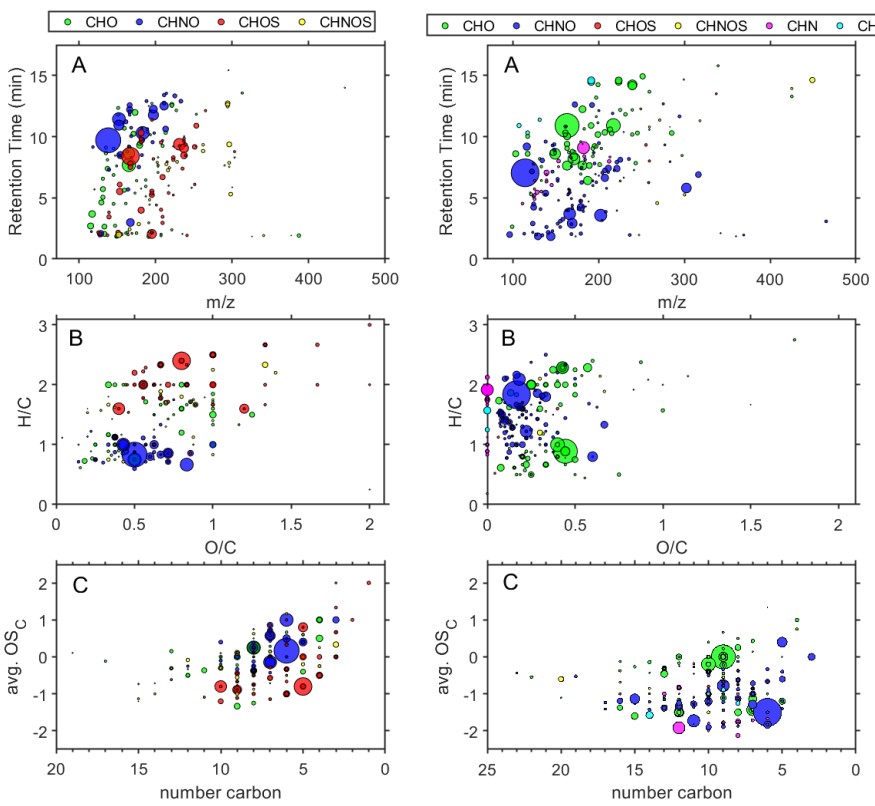

Figure 1. (A) Scatter plot of HPLC Retention time (min) vs *m/z* ratio; (B) van Krevelen diagram of O/C

ratio vs H/C ratio; (C) Kroll diagram of number of C atoms against the average oxidation state of C; Left

panels are for negative ionization mode (−)HESI, right panels are positive ionization mode (+)HESI. Area

of the marker indicates relative intensity in the reserved extract while color denotes compositional class of

the assigned compound: Green for CHO, blue for CHNO, red for CHOS, yellow for CHNOS, magenta

for CHN, cyan for CH.


Figure 1 illustrates that the dataset is indicative of a typical urban influenced WSOC profile of a

dilute sample. In the negative mode, the most significant signals are several nitrophenols and

nitroaromatics; notably $C_6H_5NO_3$ (139 *m/z*, 9.7 min, L3) and $C_6H_4N_2O_5$ (184 *m/z*, 10.3 min, L3) are



tentatively identified as 4-nitrophenol and 2,4-dinitrophenol respectively. These nitrophenols ionize

efficiently in (−)HESI which explains in part their prominence in Figure 1.B where they cluster around

0.6-0.9 H/C and are indications of biomass and fossil fuel burning emissions (Taneda et al., 2004). The

prominent CHOS compounds in the negative mode are alkylorganosulfates, notably $C_5H_{12}O_4S$ (168 *m/z*,

8.4 min, L3) and $C_8H_{18}O_4S$ (210 *m/z*, 12.9 min, L3), which are typical markers of secondary processed

automobile and shipping traffic OM.(Blair et al., 2017; Qi et al., 2021)  These can most easily be seen in

Figure 1.B above 2 H/C. Some of the other notable CHOS compounds below 2 H/C are terpene-derived

organosulfates such as camphorsulfonic acid ($C_{10}H_{16}O_4S$, 232 *m/z*, 9.4 min, L2), which also demonstrate

secondary processing under urban conditions (Iinuma et al., 2007; Surratt et al., 2007).

        The most significant positive mode signals in Figure 1 come from caprolactam ($C_6H_{11}NO$, 114

*m/z*, 7.0 min, L2) and several coumarin derivatives ($C_9H_6O_2$, 146 *m/z* 7.4 min, L2; $C_9H_8O_4$, 162 *m/z*, 10.9

min, L3; $C_9H_8O_3$, 164 *m/z*, 8.7 min, L3; $C_{10}H_8O_4$, 192 *m/z*, 9.0 min, L3). Caprolactam is a cyclic amide

and indicative of industrial emission influence as it is primarily used for manufacturing synthetic fibers

but also used in numerous other manufacturing activities. Caprolactam is a monitored compound on the

hazardous air pollutants list by the United States Environmental Protection Agency (U. S. Environmental

Protection Agency, n.d.). Coumarin species are known brown carbon components and have biomass

burning sources as well as potential secondary pathways (Xing et al., 2023).

        The several other prominent CHNO compounds are mostly amines, e.g. $C_{11}H_{23}NO_2$ (201 *m/z*, 3.6

min, L3), $C_9H_{11}NO_2$ (165 *m/z*, 3.7 min, L3), DL-Stachydrine ($C_7H_{13}NO_2$, 143 *m/z*, 1.8 min, L2), etc. This

is consistent with known amine-nitrate aerosol formation during winter months where there are sources of

amine salts and semi-volatile organic amine compounds, particularly in areas with high agricultural and

combustion emissions (Price et al., 2016). The other prominent CHNO compounds are tentatively

identified as amides such as $C_{12}H_{18}N_2O$ (206 *m/z*, 6.6 min, L3), $C_{10}H_{14}N_2O$ (178 *m/z*, 4.2 min, L4),

$C_3H_4N_4O_2$ (128 *m/z*, 1.8 min, L3), etc; which can either be further secondary products of AA (Price et al.,

2014) or the result of anthropogenic emissions (Li et al., 2022; Schollée et al., 2017). These amine and



amides tend to have lower retention times and can been seen in the lower cluster in Figure 1.A. The CHO

species present are generally either aromatics or aliphatic acids and separate out as so in the van Krevelen

diagram in Figure 1.B, with aromatics below 1.5 H/C and acids above such as $C_{12}H_{24}O_3$ (216 $m/z$, 14.2

min, L3) and $C_9H_{10}O_3$ (166 $m/z$, 9.1 min, L3). Very few biogenic CHO species are present as there are

very few CHO species within 1.5-1.8 H/C that would indicate humics, ligins, or other raw biomass

markers (Qian et al., 2013). This is consistent with the winter season sampling. Further characterization of

the nonaromatic CHO is difficult to generalize, as there are a variety of sugars, ethers, alcohols, and acids

that represent various possible biogenic species, terpenoids, and terpene derivatives. For example, xylitol

(possibly arabitol) ($C_5H_{12}O_5$, 152 $m/z$, 1.6 min, L3), hexitol ($C_6H_{14}O_6$, 182 $m/z$, 1.6 min, L3), cinnamic

acid ($C_9H_8O_2$, 148 $m/z$, 8.7 min, L3), phthalates such as dimethyl phthalate ($C_{10}H_{10}O_4$, 194 $m/z$, 10.9 min,

L2) and phthalic acid ($C_8H_6O_4$, 166 $m/z$, 7.6 min, L3), as well as succinic acid ($C_4H_2O_3$, 98 $m/z$, 2.6 min,

L3), levoglucosan ($C_6H_{10}O_5$, 162 $m/z$ , 4.5 min, L2) and farnesol ($C_{15}H_{26}O$, 222 $m/z$, 14.6 min, L3) are all

species potentially identified in the dataset.

A few CH and CHN compounds were found only in the positive mode, primarily AA and a couple

of aromatic hydrocarbons. Combined, these represent less than 11% of the positive mode compounds.

Only one CHS species was identified ($C_{18}H_{12}S$, 260 $m/z$, 10.0 min, L4) but was not used for analysis as it

was below peak quality requirements. No CH, CHN, or CHS species were found in the negative mode.

3.2 Retention Coefficients

Table 1. Mean and Median Retention Coefficients by Compound Class and Heteroatom Group

|  | (−)HESI | | | | | (+)HESI | | | | |
|---|---|---|---|---|---|---|---|---|---|---|
|  | Mean | σ | Median | n | n% | Mean | σ | Median | n | n% |
| Total | 0.95 | 0.21 | 0.96 | 208 | 100 | 0.95 | 0.53 | 0.93 | 250 | 100 |
| CHO | 0.90 | 0.25 | 0.91 | 68 | 32.7 | 1.01 | 0.82 | 0.90 | 73 | 29.2 |
| CHN | - | - | - | - | - | 1.07 | 0.21 | 1.04 | 22 | 8.8 |
| CHNO | 0.96 | 0.08 | 0.95 | 46 | 22.1 | 0.94 | 0.21 | 0.93 | 108 | 43.2 |





| | | | | | | | | | | |
|---|---|---|---|---|---|---|---|---|---|---|
| CHNOS | 0.97 | 0.23 | 0.98 | 26 | 12.5 | 1.11 | 0.34 | 0.99 | 7 | 2.8 |
| CHOS | 0.99 | 0.23 | 0.98 | 56 | 26.9 | 0.67 | 0.59 | 0.99 | 3 | 1.2 |
| CH | - | - | - | - | - | 0.66 | 0.55 | 0.78 | 5 | 2.0 |
| O Containing | 0.95 | 0.21 | 0.96 | 196 | 94.2 | 0.97 | 0.54 | 0.92 | 191 | 76.4 |
| N Containing | 0.96 | 0.15 | 0.96 | 73 | 35.1 | 0.97 | 0.22 | 0.95 | 137 | 54.8 |
| S Containing | 0.99 | 0.23 | 0.98 | 83 | 39.9 | 0.98 | 0.44 | 0.99 | 10 | 4.0 |


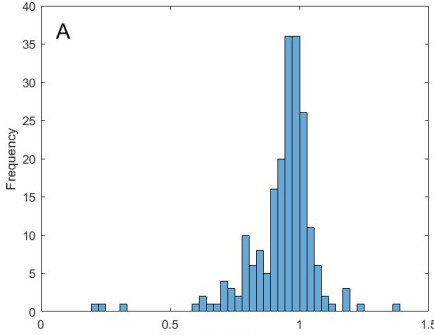
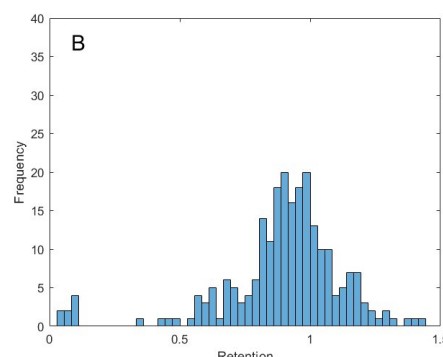

Figure 2. Histograms of Retention Coefficients for all measured L5 and above compounds that met peak quality constraints; (A) (−)HESI, (B) (+)HESI






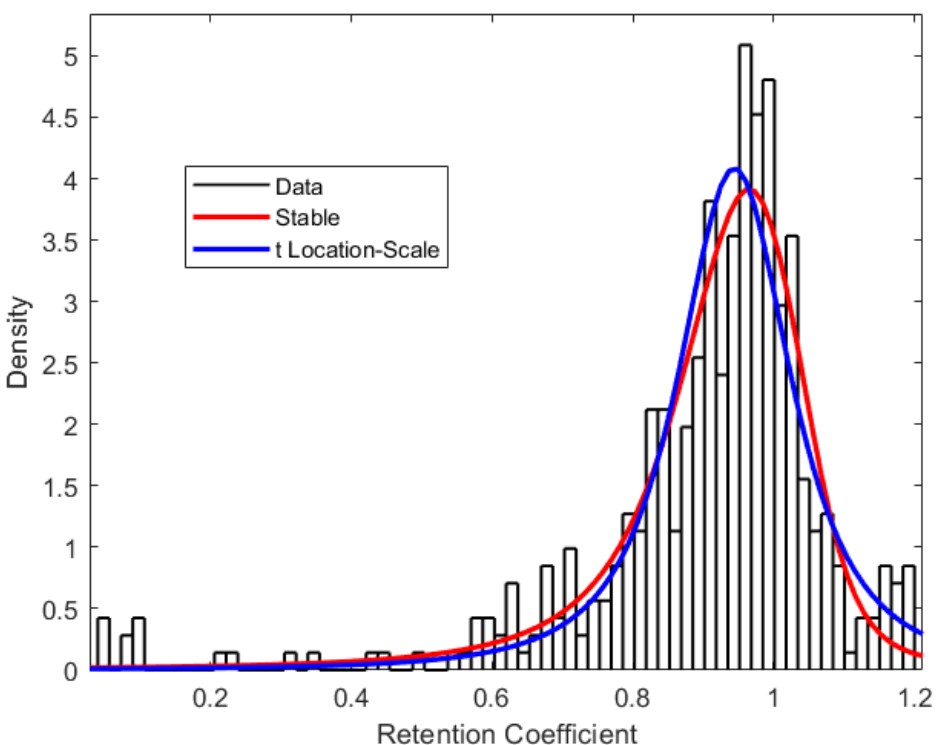

Figure 3. Merged Histogram for (−)HESI and (+)HESI of all Retention Coefficients measured fit with Stable and t Location-Scale distribution parameterizations of statistical density.

Table 2. Parameters for Stable Distribution Fit presented in Figure 3

| Mean | 0.8894 | |
|---|---|---|
| Log Likelihood | 224.657 | |
| Parameter | Value | Std. Error |
| Alpha (α) | 1.38642 | 0.06988 |
| Beta (β) | -0.61652 | 0.10241 |
| Gamma (γ) | 0.07289 | 0.00383 |
| Delta (δ) | 0.95409 | 0.00608 |


Table 3. Parameters for t Location-Scale Distribution Fit presented in Figure 3

| Mean | 0.9442 | |
|---|---|---|
| Log Likelihood | 210.775 | |
| Parameter | Value | Std. Error |
| Mu (μ) | 0.94415 | 0.00566 |
| Sigma (σ) | 0.08665 | 0.00630 |
| Nu (ν) | 2.02867 | 0.26969 |

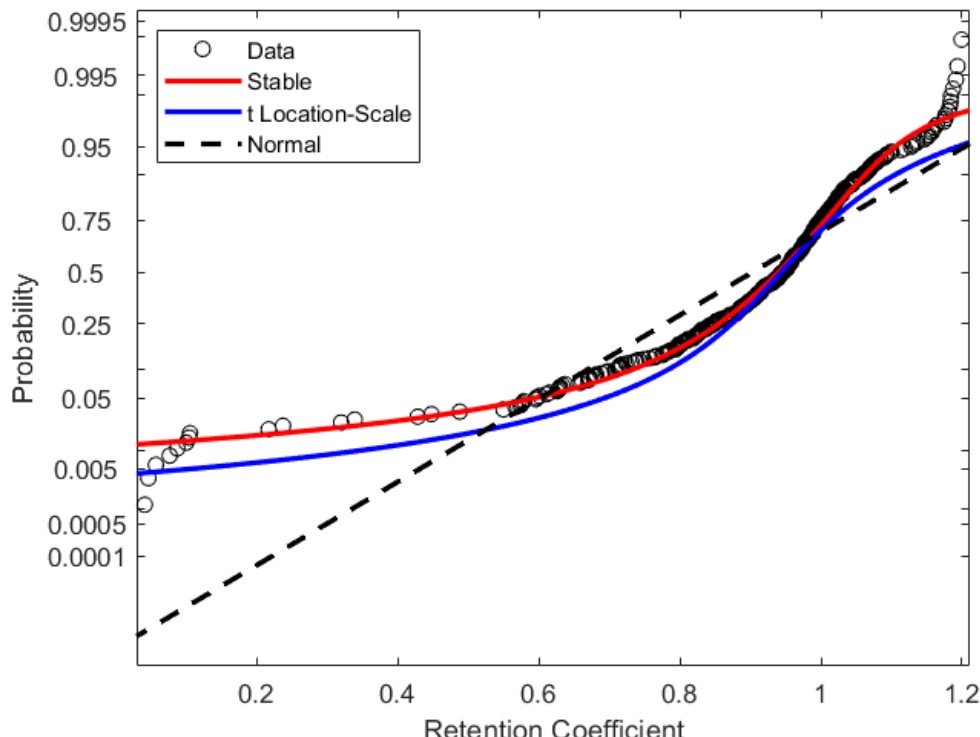

Figure 4. Probability plot of Retention Coefficients with the Stable and t Location-Scale distribution
parameterizations and a Normal distribution

The histograms presented in Figure 2 and Figure 3 illustrate the distribution of retention

coefficients determined for this dataset. Each histogram shows a peak at 0.96, 0.93, and 0.94 for (−)HESI,

(+)HESI, and the full dataset respectively with average retentions all at 0.95 with standard deviations of

0.21 and 0.53 respectively. These values and the values for each compound class are presented in Table 1.

Visually the distributions in Figure 2 appear nonnormal, suggesting a true distribution is being measured.

Additionally, Figure 4 shows the data deviates strongly from a normal distribution. Both Shapiro-Wilk

and Shapiro-Francia tests indicate nonnormality (p-values: 0.4052, 0.3940 (−)HESI; 0.5698, 0.5611

(+)HESI respectively).





320        Combining these distributions and filtering out outliers, the dataset is fitted with two distributions

to model the dataset: Stable and t Location-Scale. The parameters for these fits can be found in Tables 2

and 3. These distributions are functional estimations of the statistical density of retention coefficients

based on the empirical measurements in this experiment. The Stable distribution appears to model the

data most accurately as the parameter errors are lower than the t Location-Scale and the data points in

Figure 4 lie closer to the Stable distribution curve.

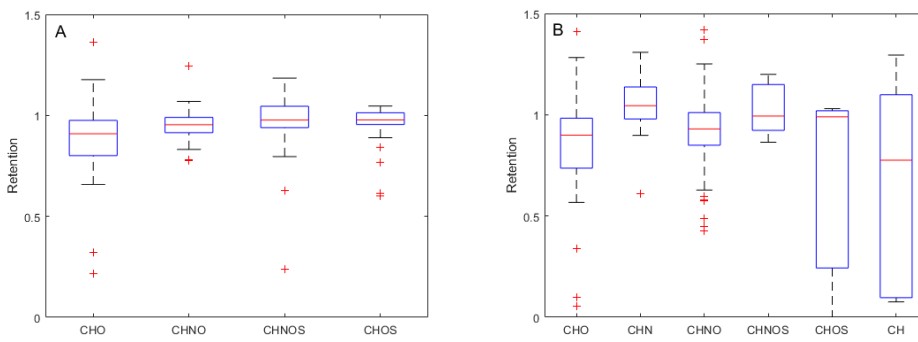

Figure 5. Boxplot of Retention Coefficients by Composition Class; (A) (−)HESI, (B) (+)HESI

330        The means of the composition classes in (−)HESI vary little, generally less than 5% from each

other. In contrast, the means of the composition classes in (+)HESI vary more, up to 40%. The deviations

and ranges of values are also wider in (+)HESI, from 0.21 to 0.82 as seen in Table 1. Visually this can be

seen in Figure 5. For both ionizations, CHNOS tends to be the highest retained along with CHN in

(+)HESI. CHNOS represents more of the heaviest species in the sample set while CHN is entirely AA. In

(+)HESI, CHNOS, CHOS, and CH represent the smallest portion of the dataset—less than 6%—and

some of the most variably retained species. For CH, this follows with the variability in hydrocarbon

aqueous solubility, however this variability is more likely explained for CHNOS, CHOS as well with the



smaller sample set bias. Notably, CHO has lower retention with a wider distribution than CHNO in both

ionizations. In (−)HESI, CHNO is mostly nitroaromatics while in (+)HESI, CHNO is mostly amines and

amides. CHO represents a more similar distribution of organic acids and terpenoids in both positive and

negative mode, with more nonpolar species represented in (+)HESI. The lower retention among CHO

may then be based on its distribution of organic acids versus terpenoids. This data suggests that nitrate

species and amines/amides have similar retentions. It is known that $NO_x$ removal is enhanced by aqueous

phase reactions (Daito et al., 2000) and that organic nitrogen represents an important fraction of WSOC

(Saxena and Hildemann, 1996; Zhang et al., 2002), but this data may also indicate that nitrogen chemistry

on CHO species enhances their retention in hydrometeors. Other UHRMS studies of rainwater have

suggested similar explanations for nitrogen uptake in hydrometeors and rainwater organic nitrogen's high

bioavailability (Seymore et al., 2023).

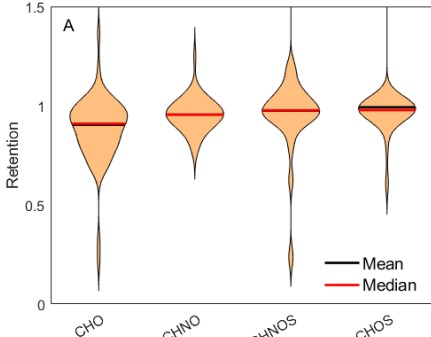 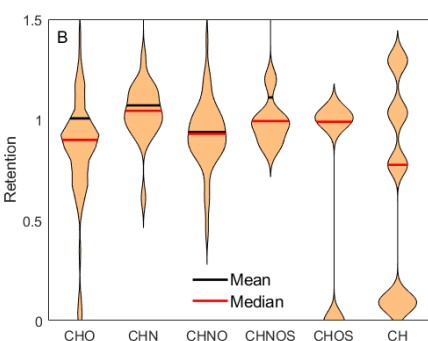


Figure 6. Violin plots of Retention Coefficients by Composition Class; (A) (−)HESI, (B) (+)HESI

        The distributions of CHNO and CHOS in (−)HESI seen in Figure 6 show a coincidence of the

mean and median along with strong symmetry around the mean. This indicates visually that they appear

to be normally distributed, suggesting that the true retentions for the whole of these compound classes

may be close to 1. Both Shapiro-Wilk and Shapiro-Francia tests indicate normality (p-values: 0.1920,



0.1922) for CHNO but not for CHOS ($7.7595 \times 10^{-9}$). It therefore would be reasonable to assume a normal distribution for CHNO, but CHOS cannot be rigorously stated as normally distributed. CHO and CHNOS show unique distributions with a significant number of values within -σ and below, indicating there are certain CHO species that are not retained during freezing.

For (+)HESI, the samples sizes for CHNOS, CHOS, and CH are too small to make meaningful descriptions of their distributions. For CHO, CHN, CHNO, the distributions are visually nonnormal and also do not pass any statistical test for normality. The distributions also appear less smooth than their negative mode counterparts, likely a result of previously discussed ionization variability in (+)HESI. Notably, (+)HESI shows a few species with very low retention specifically within the CHO, CHOS, and

CH groups. Specifically, these are $C_{14}H_{22}$ (190 *m/z*, 14.5 min, L3), $C_{15}H_{26}O$ (222 *m/z*, 14.6 min, L3), and what appears to be a phenyl-sulfide species ($C_{16}H_{18}OS$, 258 *m/z*, 8.0 min, L4). This is the only identified organosulfide within the dataset. It would be speculative to say that its low retention may indicate that organosulfides as a class are unretained and thus unlikely to appear in the dataset. Its low retention likely has more to do with its low polarity. $C_{14}H_{22}$ and $C_{15}H_{26}O$ as long chain, nonpolar species demonstrate that

species with lower aqueous solubilities are also likely to have low retentions.

Concerning heteroatoms, the distributions and ranges of retentions are quite similar among all groups. Oxygen-containing species appear to have a slightly wider distribution which is mostly weighted by the CHO class. Nitrogen-containing species have a smaller standard deviation than the O or S containing species, indicating fewer species with variable retentions and more fully retained compounds. This further

suggests that nitrogen inclusion enhances retention (see also Figure S1).

3.3 Correlation of Retention Coefficients with Chemical Properties





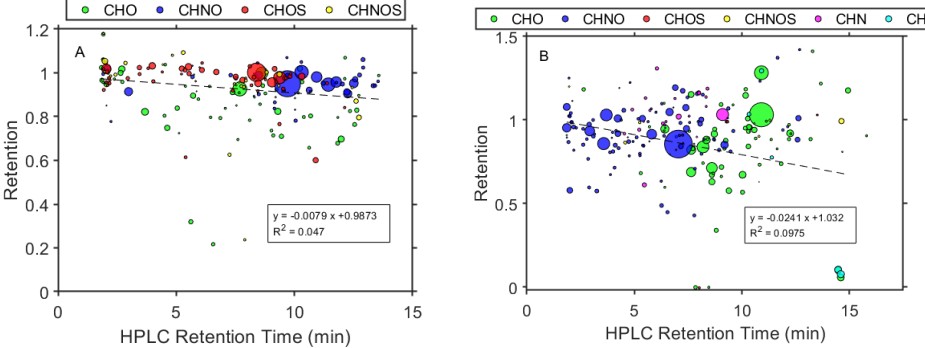


Figure 7. Retention Coefficient as a function of HPLC Retention time; (A) (−)HESI, (B) (+)HESI; Color denotes compositional class of the assigned compound, as used in Fig. 1: Green for CHO, blue for CHNO, red for CHOS, yellow for CHNOS, magenta for CHN, cyan for CH. Dashed line shows linear fit.

The determined molecular weight (MW) shows little correlation linearly with retention (as seen in Figure S2). In (−)HESI, there is a weak positive correlation, suggesting larger compounds are more likely to be retained. This is likely related to lower vapor pressures associated with larger MW species in the negative mode. An F-test against the constant value model indicates that this correlation is not significant (p-value: 0.0857). However in (+)HESI, there's a very weak negative correlation for MW

which suggests the opposite. An F-test against the constant value model indicates that this correlation with MW is also not significant (p-value: 0.1440). This trend in the positive mode is likely driven more by polarity, as Figure 1.A also demonstrates that larger species in (+)HESI tend to have higher HPLC retention times and are therefore more nonpolar. The plot in Figure 8 further demonstrates this with a stronger negative correlation between the HPLC retention time and the retention.

Figures 7 shows correlations for retention coefficients with HPLC retention time. With reverse-phase HPLC, retention time is a direct proxy for molecular polarity, i.e. shorter retention times indicate higher polarity and longer retention times indicate more nonpolar species. Both (−)HESI and (+)HESI show significant negative correlation between retention and retention time and therefore polarity; an F-





test against the constant value model shows p-values of 0.00193 for (−)HESI and $1.44 \times 10^{-6}$ for (+)HESI.

This indicates that nonpolar species are likely to be unretained and this appears to be especially true for

the previously discussed long chain species, such as $C_{14}H_{22}$ (190 *m/z*, 14.5 min, L3) and $C_{15}H_{26}O$ (222

*m/z*, 14.6 min, L3) compounds. In Figure 7, a few compound classes separate distinctly by polarity,

particularly the CHOS and CHNO in (−)HESI as well as CHO and CHNO in (+)HESI. These polarity

differences in these classes may be the driving force in the difference of the retention between CHOS and

CHNO in (−)HESI, but unlikely for CHO and CHNO in (+)HESI as CHO spans a much wider range of

retentions that cannot be explained solely by polarity.

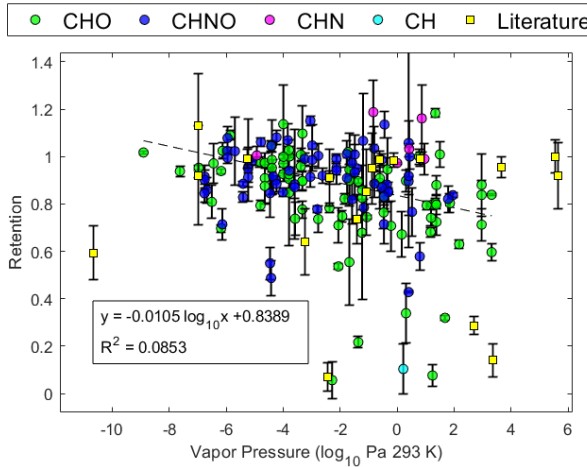

Figure 8. Retention Coefficient as a function of Estimated Vapor Pressure; Color denotes compositional

class of the assigned compound: Green for CHO, blue for CHNO, magenta for CHN, cyan for CH, yellow

squares for values taken from literature. Dashed line shows linear fit.

Further chemical property correlations with retention could be made for the species with

estimated chemical properties. The first plot, Figure 8, uses the measured retention coefficient to plot

against calculated vapor pressure (VP). It demonstrates a significant negative correlation with VP; an F-




test against the constant value model gives a p-value of $1.42 \times 10^{-4}$. It is relevant to note that the majority

of species measured are considered semi-volatile (vapor pressure: $10^{-9}$ to 10 Pa; SVOC) with few low

volatility (LVOC) and intermediate volatility organic compounds (IVOC) (Weschler and Nazaroff, 2008).

LVOC and IVOC are bias against in the sampling method as many LVOCs are highly oxygenated

compounds which may be less sensitive compared to other compounds in UHPLC-Orbitrap MS (Wang et

al., 2024) and most IVOCs are revolatilized during sampling (Bidleman et al., 2020). VP is also the

property most associated with desorption effects, likely contributing to some of the negative trend seen in

Figure 8. However, it is mostly associated with IVOC and less with SVOC so desorption alone is not

likely to fully explain this correlation.

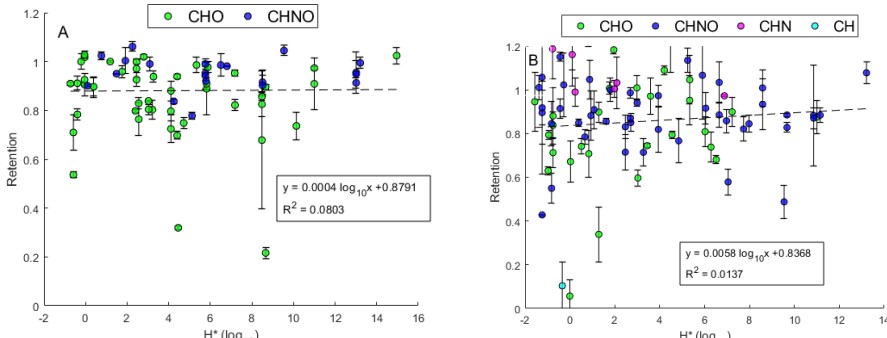


Figure 9. Effective Henry's Law Constant H* versus Retention Coefficient; (A) (−)HESI, (B) (+)HESI;

Color denotes compositional class of the assigned compound: Green for CHO, blue for CHNO, magenta

for CHN, cyan for CH. Dashed line shows linear fit.

Demonstrated in Figure 9, retention shows little or no dependency on H* under the present

experimental conditions. The linear correlations are not significant for neither (−)HESI or (+)HESI; with

p-values of 0.9270 and 0.3530 respectively for the F-test against the constant value model. The very slight



positive correlation shows agreement with Stuart and Jacobson's (2003, 2004) observation that high H*

species are more likely to be 100% retained but does not show sigmoidal behavior as modeled by Jost et

al., 2017 (plotted for reference in Figure 10). This could be the result of differing physical experimental

parameters, such as larger droplet size (2 mm versus 20 micron). However, directly comparing the known

literature values for retention coefficients with the observations made here does not immediately indicate

the systems are incompatible or exclude the comparison.

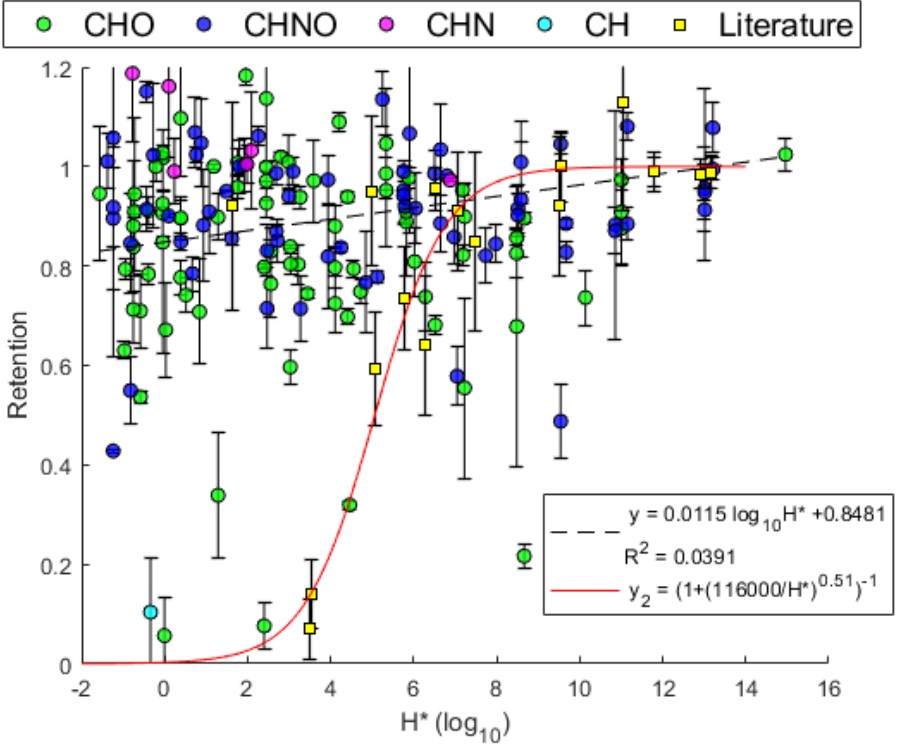

Figure 10. Effective Henry's Law Constant H* versus Retention Coefficient; Color denotes compositional

class of the assigned compound: Green for CHO, blue for CHNO, magenta for CHN, cyan for CH;

Yellow squares denote values taken from Borchers et al., (2024), Jost et al., (2017), and Von Blohn et al.,

(2011, 2013). Dashed line shows linear fit. Solid red line gives the parameterization by Jost et al., (2017).



Figure 10 compares the data presented in Figure 9 against the literature values presented by

Borchers et al., (2024), Jost et al., 2017, and Von Blohn et al., (2011, 2013). This comparison does not

appear incongruous, i.e. no discernible difference can be seen between wind tunnel experiments and these

observations. While the measurements presented by Borchers et al., (2024), Jost et al., (2017), and Von

Blohn et al., (2011, 2013) are physically dissimilar experiments to this study—i.e. wind tunnel

experiments, small droplets of micrometer size, high ventilation conditions—their observations are

congruent with this experiment. The literature values include only three species with $H* < 10^4$ and the

lowest $H*$ of them, pinandiol, is excluded from the parameterization presented by Borchers et al., (2024)

as it was perceived as an outlier. Without more measurements of compounds with $H* < 10^4$ under wind

tunnel conditions, it is difficult to determine if the nonsigmoidal behavior seen in this experiment is the

result of different physical parameters—specifically the lower surface-to-volume ratio—or if the

sigmoidal behavior described by Borchers et al., (2024) and Jost et al., (2017) is an overfit of a limited

dataset. Evidence presented by Gautam et al., (2024) in part 1 of this publication series makes a

compelling case for the dominance of physical parameters at these drop sizes. Critical to these

experiments, Gautam et al., (2024) observed the formation of an ice shell, which inhibited any further

expulsion of dissolved substances during freezing.


## 4 Conclusions

This study presents the measurement of the retention coefficients for real, complex WSOCs from

urban particulate matter for direct drop freezing under raindrop size conditions. The overall distribution of

the retention of WSOCs forms a real, nonnormal distribution up to 1. Looking at the individual

compound classes of organics, the data shows that they may have different distributions of

retention coefficients. Most negatively ionizable CHNO and CHOS compounds appear to be fully

retained, indicating that nitroaromatics and organosulphates are favorable to be retained. Slight positive



correlations between MW, polarity, and H* are seen with retention along with a negative correlation with VP. No sigmoidal relationship with H* was observed. This is likely the result of the lower surface-to-area

ratio for this drop size and the ice shell formation observed by Gautam et al., (2024) in part 1 of this publication series. However without further measurements of single component solutions for compounds with H* < $10^4$ under wind tunnel conditions, specifically for small cloud droplet sizes, it is difficult to determine if the nonsigmoidal behavior seen in this experiment is solely the result of physical parameters or if the sigmoidal behavior described by other studies is an overfit of a limited dataset.

Sulfides, lipids, aromatic hydrocarbons, and long chain compounds are among the most unretained and incidentally the fewest species observed. These are also among the most nonpolar species observed, which is presumably the dominant factor in that regard. CHO species show the highest variability for their measured retentions, most likely related to the distributions of polarity and VPs among the sugars, organic acids, and terpenoids seen here.

AA don't follow the trends associated with polarity and VP but are among the most highly retained species. The explanation for this is possibly in its structural properties, which cannot be easily determined using this analytical method. AA solubility in water is largely determined by the dimension and structure of the alkyl substituents, such that AA with longer chains are less soluble than AA with shorter chains and AA with branched substituents are less soluble than AA with linear groups with the

same number of carbons (Badocco et al., 2015). Polarity and hydrogen bonding are also known contributors to AA solubility but this is not unique to AA. Hydrogen bonding potential may enhance retention along oxygen functionalities such as among nitro and sulphate species. Hydrogen bonding alone, however, is unlikely to fully explain the high retentions among nitro and sulphate species. Specifically, nitro species have weak hydrogen bonding potential (Shugrue et al., 2016) and as a result is

likely influenced mostly by the increased polarity imparted by the nitro substituent or its dissociation.

   The data suggests that nitrogen and sulfur inclusion generally increase a species' ability to be retained. Overall, this insulates that the products of $NO_x$ and $SO_x$ reactions from anthropogenic emissions



enhances the retention of these SOA species, reducing the likelihood of reaching the upper atmosphere.

Further on this, other studies have demonstrated that $NO_x$ removal is enhanced by aqueous phase

reactions (Daito et al., 2000). These findings may also indicate that this $NO_x$ chemistry on CHO species

enhances their retention in hydrometeors, potentially by increasing its polarity or solubility. Other

UHRMS studies of rainwater have suggested similar explanations for nitrogen uptake in hydrometeors

and rainwater organic nitrogen's high bioavailability (Seymore et al., 2023). Additionally, correlations

with VP and polarity show that lower VP species and more polar species tend to be retained. Atmospheric

chemical processing generally tends to oxidatively degrade large nonpolar species into more water-

soluble, less volatile species (Iavorivska et al., 2016). Specifically, aqueous phase droplet chemistry is

known to facilitate condensed phase SOA formation from highly volatile species (McNeill, 2015).

However, many freshly aged terpene products increase in volatility or see only small decreases in VP with

oxidation for their early generation products (Bilde and Pandis, 2001; Kurtén et al., 2018; Wu et al.,

2021). This insulates that many freshly oxidized SOA precursors may have a lower potential to be

retained than aged organics and may generally suggest that freshly oxidized SOA precursors are more

likely to reach the upper atmosphere than primary organics or aged SOA.

The use of UHPLC-HRMS has allowed for the study of ambient WSOC retention rather than

single component or limited mixture experiments from previous studies. The experiment in this paper

demonstrates the viability of UHPLC-HRMS analysis for ambient WSOC and shows the need for further

complex mixture study regarding retention. Future study on retention within hydrometeors should include

complex mixture analysis under the physical conditions most similar to the atmosphere, i.e. wind tunnel

experiments, smaller droplets, increased ventilation. These studies would also be improved with more

distinguishable tracers with known retentions and more sophisticated corrections for desorption. Further

studies on single component solutions of species with $H^* < 10^4$ under atmospherically similar physical

conditions would also allow for stronger conclusions from the comparison of the retentions measured in

this study with other experiments performed under wind tunnel conditions.

The experiment presented here also cannot distinguish between species incorporated within ice crystal structure and those phase-separated but physically constrained to the hydrometeor, potentially

between crystal grain boundaries or on the particle surface. That distinction is also not atmospherically relevant regarding the net transport of organics into the upper troposphere. While this method aims to demonstrate the retention for real WSOC, this method is still sample method biased against higher volatility species and likely features other sampling bias typical for HiVol filter based measurements such as filter extraction bias and solvation effects. These measurements also present the distribution of

retention coefficients for the variety of species present and not necessarily the mass distribution of species potentially present in the atmosphere. Corrections for species abundancy must first be made in order to apply this data to organic transport models.

## Acknowledgements

Special thanks to Konstantin Dörholt for his initial work and experimentation in the Mainz Wind Tunnel with these samples.

This work was funded by the Deutsche Forschungsgemeinschaft (DFG, German Research Foundation) – TRR 301 – Project-ID 428312742.

This work was supported by the Max Planck Graduate Center with the Johannes Gutenberg University of

Mainz (MPGC).

## Author Contributions

JS, MG, MS, AT participated in designing the experiments; LZ provided the samples; JS, JM prepared the samples for experiments; MG performed the experiments; JS, JM, AV conducted the analytical



measurements; JS analyzed the data and wrote the manuscript draft; MG, MS, AT, JM, AV, TH reviewed

and edited the manuscript.

## Competing Interests

The contact author has declared that none of the authors has any competing interests.

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
