# Peer review of "Retention During Freezing of Raindrops, Part II: Investigation of Ambient Organics from Beijing Urban Aerosol Samples"

_EGUsphere, 2024_

## Author Comment (AC1)

We would like to thank the Editorial Support team for giving us the opportunity to respond to the reviewers' comments in open discussion. We greatly appreciate the reviewer's careful reading and review of this manuscript.  We have addressed each reviewer's comments and suggestions, as shown in our responses below.  Specific references to the manuscript have been highlighted in yellow and rewritten sections are provided directly below in italics for convenience. The lines, figures, and sections referenced match the numbering in the version of the manuscript presently uploaded for discussion. A revised version of the manuscript will be uploaded for the handling editor's consideration.

**Reviewer(s)' Comments to Author:**

**RC1**:

**The manuscript is based on extensive laboratory studies on the mechanism of retention/release of semi-volatile organic compounds upon freezing of raindrop-sized droplets generated from the aqueous extracts of Beijing wintertime urban aerosol. The whole set of experiment is carefully designed and executed, using state-of-the-art equipment including analytical techniques, and the statistical processing of analytical data is adequate meeting all standards of science. The objectives of the manuscript are clear, the hypotheses are valid and important, to explain a potential transport mechanism of semi-volatile organic compounds to the free troposphere. This is relevant in understanding new particle formation in the upper troposphere, which has serious implication on cloud formation and water-particle interactions in a changing climate hosting more water vapour in the atmosphere. The manuscript is comprehensive and well-written, so there is little to criticize expect typography (e.g. using 'en dash' characters instead of the 'minus sign' in all formulae.) However, the reviewer has a series of serious concerns about the relevance of the laboratory results for real-life atmosphere.**

We would like to thank the reviewer for the constructive comments. The typography issues concerning the use of 'en dash' characters and 'minus sign' characters have been addressed where found. We request the editorial team review our compliance with the journal's formatting guidelines on this issue as appropriate.

- **In the study water-soluble organic compounds (WSOC) in the aqueous extract of wintertime urban aerosol serve as a proxy for real-life composition of droplets of convective clouds. However, large-scale convective cloud formation is not typical during the winter, furthermore, in winter frequent inversion and low mixing layer height prevent surface emissions to be transported to higher**

**altitudes and participate in ice cloud formation. Whereas I agree that studying such a complex mixture may be more informative that of a few cherry-picked model species, to draw meaningful conclusions from the experiments it is important to elaborate on this issue in the manuscript.**

We have incorporated discussion of this relevant concern in ==section 4, line 511.== We would like to address that our filter samples were chosen to maximize mass loading, which was mainly achieved in the winter months.

> *It is also important to note that large-scale convective cloud formation where freezing retention is expected is not typical during the winter such as when the aerosols in the present study were sampled. Furthermore, in winter, frequent inversion and low mixing layer height tend to prevent surface emissions from being transported to higher altitudes and participate in ice cloud formation, so the compounds presented here may not be wholly representative of the species that participate in the retention process.*

- **The laboratory experiment is designed to study the freezing of large (2 mm in diameter) raindrops. In real-life mixed phase clouds freezing of large supercooled droplets may occur via multiple mechanisms, such as riming, immersion or contact freezing. These processes are also active for much smaller cloud droplets, i.e. for those below or around the precipitation threshold. I wonder what the probability is for such a large (supercooled) raindrop to survive such effective freezing processes inside a vigorous mixed phase cloud high in the troposphere? I would guess it is very low, but it would be worth discussing anyway.**

For a large, supercooled raindrop to survive such conditions is low. This might be possible in a thunderstorm cloud, but more relevantly, the chosen drop size is rather an experimental constraint. First, a relatively large liquid mass is required for chemical analysis. Second, in order to properly determine the freezing temperature, a certain drop diameter is needed. So in order to ensure viable measurements, we were required to scale the process to the larger drop sizes one might find in the atmosphere. For small drop sizes, freezing is less relevant than riming.

Riming is established as the dominate pathway for ice production in convective systems, but hetero/homogenous freezing is still an understudied contributing pathway. Previous studies have already begun to examine retention during riming. These experiments use large wind tunnel facilities to simulate rime growth. One of the objectives of this study is to establish untargeted complex mixture analysis as a method to measure retention before

we take it to the wind tunnel. We have future experiments planned as a separate publication involving the Mainz wind tunnel facility to more closely examine rime growth.

- **In addition, the real-life freezing processes described above are superfast relative to the cooling process applied by the authors in their experiments, which lasts on average for 90 seconds. So again, the question arises how relevant the experimental parameters are for real-life cloud conditions in determining the gas-to-droplet partitioning of organic compounds?**

While the total freezing was approximately 90 seconds, the most relevant freezing is likely the ice shell formation, which is very quick, on the order of milliseconds. Freezing time as a relevant physical parameter for the freezing process is addressed more in-depth in the Part I publication. https://doi.org/10.5194/egusphere-2024-3917

- **Another issue to be clarified is the application of the standing ultrasonic wave for levitating the droplets in the experimental setup. It is well-established that ultrasonic energy is an extremely effective way of mixing (see e.g. ultrasonic bath for extraction). So in terms of fluid dynamics, can we assume that in the laboratory experiments the droplets had remained thoroughly mixed until they froze up? If so, how it relates to fluid dynamics prevalent for droplets in convective clouds, in which mixing inside such large droplets may be way much less effective?**

We do assume that the ultrasonic field produces droplets that are well-mixed. Before they stabilize, the drops perform several visible oscillations that produce mixing. There is a more thorough investigation of drop mixing in the acoustic trap in Szakáll et al., JAS, 2009 (https://doi.org/10.1175/2008JAS2777.1). However, we would also say that for a 2 mm raindrop in the atmosphere that falls with 6-7 m/s, turbulent mixing will occur inside the drop. While mixing should be higher in an ultrasonic field, mixing inside a freely falling drop is also high and therefore it is justifiable that our drops are also well mixed before freezing.

Ensuring homogeneity inside the droplet is also important for maintaining reproducibility for this experiment. The calculations we use for determining freezing retention assume the droplet is at chemical equilibrium before freezing. Nonhomgenous conditions inside the droplet would invalidate these assumptions. However, we estimate that it would not produce significantly different results specifically for freezing retention so long as diffusive influence could be exactly controlled and accounted. While nonhomogenous mixing of large droplets would relevantly influence diffusive transport as it creates a concentration gradient, its influence on the freezing retention should be negligible. As a phase separation process, unless a species is near to saturation concentrations (unlikely in naturally

occurring scenarios and would likely be insoluble and unretained to begin with), the freezing retention has not been demonstrated to be concentration dependent.

- **The last issue is that in large droplets mass transfer to the atmosphere is strongly limited by the low surface area to mass ratio. Furthermore, freezing of the droplets starts from the outside because the enthalpy of fusion needs to be dissipated, then freezing forms a solid outer layer that hinders heat and prevents material transport to and from the interior of the droplet. I wonder if this mechanism is largely responsible for the nearly complete retainment of soluble species irrespective of the vast range of their physical and chemical properties?**

We appreciate the reviewer's good observation. This 'ice shell' formation is discussed is addressed more in-depth in the Part I publication. https://doi.org/10.5194/egusphere-2024-3917

We would like to add that the retention can also be understood as the ratio of the chemical expulsion timescale and the freezing time (or ice shell formation time).

**To summarise my general comments it would be more than welcome if the authors addressed these points in their revised manuscript.**

We would like to thank the reviewer for their time and their contribution of constructive comments.

**RC2:**

**This study presents the retention of organics during the freezing of rain size droplets. The experiments were conducted on aqueous extracts of Beijing urban ambient aerosols using an acoustic levitator. The retention coefficients were determined by freezing the rain size droplets and then measured the remained substances in the frozen droplets. They showed near 1 of retention coefficients for the compounds found in the HRMS in (−)HESI mode, but more variation for the compounds found in (+)HESI mode. Correlations between the estimated chemical properties, such as henry's low constant, and retention coefficients were discussed. The subject of this manuscript fits the scope of ACP. There are several issues need to be addressed before it can be considered for publication.**

We would like to thank the reviewer for the constructive comments.

**Major comments:**

1. **The manuscript presented in the current form is more like a Measurement Report, as its atmospheric implication is not fully discussed or addressed. The only three nighttime samples also limits a broader atmospheric implication. If I understood correctly, the samples were combined before the freezing and determination of retention coefficient. Why not perform the experiments for individual samples, which one can exam the variation of retention for the same compounds?**

There are a variety of reasons why we chose to combine filter samples for our experiment. Using combined samples instead of individual samples eliminates variability to allow for experimental reproducibility and valid statistical evaluation of the retention coefficients. Specifically, some of these confounding variables that are controlled by this method could be large concentration differences between samples, background matrix influences, pH, biasing of trace species, etc. The individual samples on their own also do not produce adequate mass loadings for experimental viability. Combining sample extracts allows for enough liquid volume to be produced for the experiment without unduly diluting the sample. Further, comparative analysis of the individual filter samples is outside objectives of this publication and would also require examining the synoptic conditions during filter sample collection.

2. **In the Abstract, the statement in Line 20-22 is overstated as the S- and N-containing compounds are not solely from the NOx and Sox chemistry. The statement in Line 24-26 is not supported by the results of non-sigmoidal relationship or the related discussion. The lower surface-to-volume ratio of the large drop size investigated may be one of the reasons. The differences in the experimental techniques, as I understand, may be the main reason, the wind tunnel and levitation, which have totally different air dynamic environments and surrounding corresponding gas-phase organic concentrations that will affect the gas-liquid diffusion and partitioning, and thus the retention.**

We have amended lines 20-22 to more accurately state what compounds are being referenced. We have amended lines 24-26 to more precisely state the confidence of correlation. We agree that the differing physical parameters of this experiment is a significant barrier to its comparison with wind tunnel studies; we present discussion on this at the end of section 3.3 and in the conclusions in lines 469-474. One of the objectives of this study is to establish untargeted complex mixture analysis as a method to measure retention before we make more direct comparisons to the wind tunnel. We have future experiments planned as a separate publication involving wind tunnel experiments. With

interest for this discussion, our preliminary data from the tunnel shows similar results to this experiment in the levitator. This gives us reason to believe that the physical differences of these experiments is not the cause for the absence of sigmoidal behavior here.

> *The findings here also indicate that N- and S- containing compounds, likely secondary organic aerosols (SOA) anthropogenically related to $NO_x$ and $SO_x$ chemistry, have enhanced retention. An insignificant positive correlation between polarity and freezing retention along with a significant negative correlation with vapor pressure and freezing retention was observed.*

3. **The limitation of this technique or the limited number of sample size should be discussed. For example, as the authors indicate the implication for the convective clouds, will the values measured by this levitation techniques be likely underestimated for the convection system?**

We have updated line 120 to highlight the actual number of drop samples that were collected and analyzed from this experiment (30 drops). We have expanded our discussion in section 4, particularly lines 512-517 to better address the limitations and implications of this study to convective clouds.

> *Enough drops to reach the minimum viable sample volume for analysis, 50 μl, were collected to produce a single sample (approximately 12 drops). 30 drops were produced in total, which allowed for two full samples to be used for UHPLC-HRMS analysis.*

> *It is also important to note that large-scale convective cloud formation where freezing retention is expected is not typical during the winter such as when the aerosols in the present study were sampled. Furthermore, in winter, frequent inversion and low mixing layer height tend to prevent surface emissions from being transported to higher altitudes and participate in ice cloud formation, so the compounds presented here may not be wholly representative of the species that participate in the retention process.*

4. **The correlation of retention coefficients with chemical properties. The discussion on the results of these correlations should be carefully examined. For example, Line385-392, the R square values are less than 0.1 and the F-test shows no significance, why do the statements still say that they have correlations? Then, the related conclusions are not valid anymore.**

For the precision of statistical discussion, an insignificant correlation is still considered a correlation. To state that there is no correlation is to demonstrate that the population correlation is zero. Our failure to reject the null hypothesis however does not confirm the null hypothesis. The r-squared and F-test indicate that there is no practically important linear relationship, not that there is no linear relationship, i.e. that population correlation is zero. The F-test indicates that we can't prove r is significantly different from zero. It doesn't prove that r is 0 with significance. However, we agree that this phrasing can be misleading, especially outside of statistical context. We have adjusted the phrasing used in throughout section 3.3, specifically lines 385-395, 411, 430, and 433 to swap some instances of 'correlation' with 'regression' as to not imply undue significance. The conclusions around the insignificant correlations found are still valid.

> *The determined molecular weight (MW) shows little correlation linearly with freezing retention (as seen in Figure S2). In (−)HESI, the slope of the regression is weakly positive, suggesting larger compounds are more likely to be retained. This is likely related to lower vapor pressures associated with larger MW species in the negative mode. An F-test against the constant value model indicates that this correlation is not significant (p-value: 0.0857). However in (+)HESI, the slope of the regression is weakly negative for MW which suggests the opposite. An F-test against the constant value model indicates that this correlation with MW is also not significant (p-value: 0.1440). This trend in the positive mode is likely driven more by polarity, as Figure 1.A also demonstrates that larger species in (+)HESI tend to have higher HPLC retention times and are therefore more nonpolar. The plot in Figure 7 further demonstrates this with a stronger negative correlation between the HPLC retention time and the freezing retention*

**Other comments:**

1. **Line227-234, it is not clear what are those compound number means. How many compounds were used for and analysis? Does that include the 77 and 84 compounds which were selected for additional property calculation?**

The MS features described in lines 227-234 are relevant to understanding how the untargeted mass spectrometry dataset was handled. Lines 228 and 232 state directly how many compounds were used for analysis from the positive and negative modes. From those 208 and 250 compounds, the 77 and 84 were then used for the additional property calculations.

2. **Line 235-237, the sentence is confusing.**

We would like to thank the reviewer for the constructive comments. We have rewritten lines 235-237 for clarity.

> *Phosphorous containing species can represent almost a third of positively ionizable species in rainwater WSOC (Seymore et al., 2023) so these species likely make up a significant portion of species variety that is not considered.*

3. **Figure 2B, why there are such a large portion of the compounds have retention coefficients higher than 1.0?**

Retention coefficients greater than 1.0 are an indication of a certain level of experimental error. Since the retention coefficient is measured relative to a standard, any change in the signal of the standard will falsely inflate or decrease the measured retention coefficient.

4. **Line316, why a nonnormal distribution is expected or is a true distribution?**

A normal distribution may indicate that all retention coefficients are a constant value with a normal distribution of error. We expect a nonnormal distribution as this would indicate that we are observing a range of retention coefficients, as we expect from a variety of species with different H*.

5. **Line393, in the sentence, the Figure 8 should be Figure 7?**

We have corrected this typo in line 393.

6. **Line398 and Figure 8, can you comment on why the R square is so low?**

An F-test against the constant value model is a comparative test that indicates whether the proposed regression is more likely to be true than the null hypothesis, i.e. no correlation. If the F-test passes, we can state that the correlation (Pearson R) is not zero even if the $R^2$ is low. It indicates that these chemical properties are likely factors in determining retention (Pearson R is not zero), but they alone cannot fully explain retention ($R^2$ is low).

7. **Line492 and 505, the authors mean "indicates" not "insulates"?**

We have fixed these typos in lines 492 and 505.

8. **Line505, lower potential to retain, why is likely to reach the upper atmosphere?**

Species that are less retained are believed to be in the gaseous state after freezing and therefore able to make it to the convective outflow. This is in contrast to a species that is retained in the hydrometeor and then deposited in wet deposition. These freshly oxidized SOA precursors described have higher VP and therefore are less likely to be retained than aged or primary organics.

We would like to thank the reviewer for their time and their contribution of constructive comments.

**RC3:**

**This paper addresses an important phenomena potentially impacting new particle formation in the upper troposphere, retention during raindrop freezing. The work substantially expands the breadth of compounds considered previously through the use of well-described state-of-the-art methods and sophisticated analyses. Results are interesting and provide insights into differences in retention based on compound classification and functionality. They further suggest impacts of atmospheric chemical processing on retention. However, the discussion of theory-based hypotheses and rational is somewhat lacking and leaves out relevant previous work, particularly regarding the expected impacts of freezing conditions on retention of low Henry's law constant compounds. Further, the conclusions on retention of general chemical classes and the impact of NOx chemistry are overstated based on the evidence presented and on the necessary uncertainty associated with large classes of partially identified compounds. They should be more nuanced. The manuscript is generally easy to read and understandable but is sometimes repetitive and vague. It also includes too many graphics that are somewhat repetitive or not informative enough for the main text. Detailed data on the compound data informing the analysis should also be provided in a supplemental table.**

We would like to thank the reviewer for their constructive comments. The MS data the reviewer requests is too large to be meaningfully incorporated into a supplemental table and therefore we will provide it through zenodo. We have published the requested data as two .csv files on that platform. <https://doi.org/10.5281/zenodo.15166745>

**Specific comments, listed by line number (more or less in sequential order):**

**Abstract (and conclusions): The conclusions on retention are too broadly applied to entire classes of chemicals when there is not enough data specifically for that class. The least supported is for 'sulfides' when only one organosulfide was identified in the dataset. The authors acknowledge this in line 368, but still state the unsupported general conclusion in the abstract and conclusions.**

Line 20 and line 475 directly state that these species are the least observed in the dataset. To avoid unintended overstatement, these lines have been reworded to refer more specifically to the observed compounds of that class. Even then, our reasoning in line 368 still supports that the speculation of low retentions for these classes could be a reason for their underrepresentation in the dataset.

> *Most nitroaromatics and organosulfates were fully retained along with the aliphatic amines (AA) and higher-order amines and amides while the observed sulfides, lipids, aromatic hydrocarbons, and long chain compounds are among the most unretained and incidentally the fewest species present.*

**20 and 476. The fact that sulfides, lipids, aromatic hydrocarbons, and long-chain compounds were not observed in high quantities is not incidental to the conclusions of this work, as small sample size was acknowledged as likely impacting the results.**

Please see our reply to the previous comment.

**21-22. The statement that anthropogenically related NOx and SOx chemistry enhances retention of the resulting secondary organic aerosol' is confusingly written. Are the authors saying that the compounds created are more polar and less volatile, increasing the formation of/partitioning to the condensed phase of the aerosol, and also retention of component species during freezing? Clarify this.**

We have amended lines 20-22 to more accurately state what compounds are being referenced.

> *The findings here also indicate that N- and S- containing compounds, likely secondary organic aerosols (SOA) anthropogenically related to $NO_x$ and $SO_x$ chemistry, have enhanced retention.*

**23-25. The sentence is a run on sentence and uses a double negative, which is confusing.**

We have split this sentence in lines 23-25 into two separate sentences to increase clarity.

> *No sigmoidal relationship with effective Henry's law constant was observed. This differs with the parameterizations of riming retention presented in current literature, which is justified by the lower surface-to-volume ratio of the large drop size investigated.*

**105-113. A diagram of the freezing experimental setup is needed.**

The diagram and associated images of the experimental setup are more presented and more thoroughly explained in the Part I publication. https://doi.org/10.5194/egusphere-2024-3917

**156. Provide a brief reason why phosphorus containing species were not considered.**

Naturally occurring phosphorus containing compounds are present almost exclusively as organophosphates. This means that there is always at least 4 O atoms per P atom present. The heuristic filtering necessary to ensure these correct ratios while maintaining compound assignment validity was outside of the scope of the analysis. More specifically, performing accurate compound assignment for P containing compounds is difficult using the native version of CompoundDiscover. It often requires externally processing the data using MIDas or an in-house software. Newer versions of CD are making this process better and more accurate so future studies using untargeted analysis will include P containing compounds.

**156 (and elsewhere). "level 5 (L5) and higher compounds". Based on the context, I think 'higher' is referring to L1-4. This is confusing because 1-4 are lower numbers than 5. This is confusing; it would be better to just provide the L range being referenced.**

We have incorporated this into the manuscript in lines 156, 157, 165.

> *All level 5 (L5) through level 1 (L1) compounds...*

**169. "These properties" suggests more than one type of property was predicted here. However, only the effective Henry's law constant is mentioned. Where other properties predicted here or is this a typo?**

There is more than one property that has to be incidentally calculated to estimate an effective Henry's law constant; specifically vapor pressure and aqueous solubility. It is also calculated through two different methods; bond contribution or group method. For simplicity we omitted these details and were left with the grammatical error. We have amended line 169 accordingly.

> *This was predicted using the HENRYWIN$^{TM}$ model as part...*

**219-220. Provide some rationale for nonuniform desorption effects being negligible.**

Desorption is largely dependent on vapor pressure and equilibration time. Our samples are from HiVol filters, which bias for a relatively narrow range of vapor pressures.

**330. "The means of the composition classes in (-) HESI vary little, generally less than 5% from each other". This statement is vague and misleading. The range of means is**

**0.90 to 0.99 in Table 1. Just state the range rather than making a statement that could misrepresent the data.**

We have incorporated this into the manuscript in line 330.

> *The means of the composition classes in (−)HESI range between 0.90 and 0.99 with an unweighted average of 0.95 and a standard deviation between them of 3%. In contrast, the means of the composition classes in (+)HESI vary more, up to 40%.*

**338. "CHO has lower retention ...than CHNO in both ionizations". This is not entirely supported by Table 1 for the positive ionization for which CHO had mean retention of 1.01 and CHNO had a lower mean of 0.94. The median is lower for CHO, but not the mean. This needs a more precise and nuanced description.**

We have incorporated a more precise description into the manuscript in line 338.

> *Notably, with lower median retentions in both ionizations, most CHO compounds have lower retentions than CHNO compounds.*

**345-346. The conclusion that nitrogen chemistry of CHO enhances retention is not strongly supported by the data described here, given the above inconsistency (see comment on line 338), and the small differences between the means compared to the error (with error indicated by measured retention coefficients of a lot more than 1 for many compounds). This really should be stated more as a hypothesis of expected differences between composition class and what might explain them, and the degree to which the evidence supports (or doesn't support) the hypothesis.**

The previous comment to increase clarity on line 338 does not invalidate the conclusions made in lines 345-346. Both the mean and median for CHO are less than those for CHNO in the (-)HESI. While the mean is higher for CHO than CHNO in the (+)HESI, the variance is significantly larger (0.82 vs 0.21) and the median is lower (0.90 vs 0.93). While that alone gives enough information to state there are more species with lower retention in CHO than in CHNO, the visual distributions in Figures 5B and 6B also demonstrate that the lower two quartiles in CHO are significantly lower than those for CHNO.

**Figures 2, 3 and 4 and Tables 2 and 3 could be provided in supplemental materials, as they are not instrumental to the main results.**

In our opinion, Figures 2, 3, and 4 as well as Tables 2 and 3 are instrumental to demonstrating that the retention coefficients we are measuring are not simply a normal distribution of error around a constant value. Further, they are necessary for any reader who intends to incorporate these results into a model.

**Figures 5 and 6 are not both needed and could be combined into one figure (such as with an overlay, or just eliminate Figure 5)**

We believe that Figures 5 and 6 are both necessary and demonstrate independent points. Figure 5 demonstrates the differences between retentions of compound classes in a statistically familiar way. Figure 6 is to demonstrate which compound classes follow nonnormal distributions and which we believe are normal distributions around constant values.

**Supplemental material should include a detailed table of compounds used in the analyses (those supporting the tables and figures shown in the main text), including their level, masses, structures, assigned composition, properties predicted, and CAS number match (when applicable and used for properties rather than a model prediction).**

Please see our previous reply.

**375. The sulphur-containing class had the highest mean/median retentions. This should be mentioned and discussed. Line 492 later concludes that SOx reactions enhance retention, but this is not discussed with the results.**

Discussing heteroatom classes like such is not always practical, as they are composed of compound functional groups with differing properties. Especially with the S- containing species being some of the fewest species observed in the dataset, it was not useful to discuss heteroatom groups as such. The conclusion in line 492 is based on the discussion of organosulphates found in the CHOS (-)HESI from lines 252-257.

**385-391. The trends with molecular weight do not seem to warrant this much discussion given they are inconsistent. Further the reasons involve the influence of vapor pressure and polarity, and require the evidence for each of those relationships. I would make more sense to put this paragraph after those discussions (and shortening it).**

As requested, we have moved the discussion of MW paragraph from lines 385-394 to after the discussion of polarity (Lines 395-406). We maintain that this discussion of MW is relevant as it is a fundamental chemical property with direct implication for the polarity and VP of a compound. It is also a directly measured chemical property rather than an estimated property like VP or H*. It would be remiss if it was not properly discussed despite its inconsistency as a predictor of retention.

**392-3 and 396-7. These are repetitive as they give the same explanation. Perhaps discuss the relationship with polarity first, and then the explanation in 392-3 won't be necessary.**

We have moved the paragraph containing lines 392-3 to after the paragraph containing lines 396-7. Line 392-3 still requires the relation of HPLC retention times to polarity as it is not a step in reasoning that is immediately noticeable from Figure 1.A.

**Some description of the theoretical hypotheses and reasons for why retention is expected to have a relationship with these properties (through reference to previous retention or other literature) is needed.**

We believe that there is suitable general discussion and reference to literature on this topic in lines 72-74. This topic is also discussed in the Part I publication (https://doi.org/10.5194/egusphere-2024-3917).The chemical properties discussed in section 3.3 are some of the fundamental chemical properties that may determine emergent chemical properties such as H*, gas or liquid diffusivity, which are commonly discussed in the literature as the most important chemical properties for retention (Stuart and Jacobson 2003,2004; Jost et al. 2017). We feel that adding more discussion to the relevance of these fundamental chemical properties would not adequately service the reader's understanding of their importance to freezing retention, but would distract from the point that H* may not be as relevant to understanding freezing retention as previously thought.

**444-459. The discussion of the relationship with effective Henry's law constant does not adequately address the potential influence of the conditions of freezing and freezing kinetics that are expected to have important impacts on retention coefficients for species with lower effective Henry's constant, as discussed in previous literature. Although the formation of an ice shell as inhibiting expulsion is mentioned at the end here, it appears to be largely an afterthought. The data are consistent with freezing conditions that enhance trapping, increasing retention even for the lowest H* compounds (which are expected to have high variability in retention dependent on freezing conditions). The assumption that a sigmoidal shape is expected, irrespective of freezing conditions does not do the previous literature on the retention phenomena adequate justice. (See for example Stuart and Jacobson 2003, 2004 (cited in this manuscript) and 2005, doi: 10.1007/s10874-006-0948-0).**

We have discussed the influence of physical parameters on freezing retention more thoroughly in the Part I publication. https://doi.org/10.5194/egusphere-2024-3917

We agree that the differing physical parameters of this experiment is a significant barrier to its comparison with wind tunnel studies; we present discussion on this at the end of section 3.3 and in the conclusions in lines 469-474. One of the objectives of this study is to establish untargeted complex mixture analysis as a method to measure retention before we make more direct comparisons to the wind tunnel. We have future experiments planned as a separate publication involving wind tunnel experiments. With interest for this discussion, our preliminary data from the tunnel shows similar results to this experiment in the levitator. This gives us reason to believe that the physical differences of these experiments is not the cause for the absence of sigmoidal behavior here.

**453-455 (and 469-473). Although this is an appropriate limitation to discuss, it is too narrow. Why focus only on the potential effect of the surface to volume ratio rather than other conditions of freezing, when there are other factors that have been suggested previously as likely important based on theory and modeling?**

We have discussed the influence of physical parameters on freezing retention more thoroughly in the Part I publication. https://doi.org/10.5194/egusphere-2024-3917

**482- 490. Solubility is expected to influence retention but was not studied here and is only discussed as a rationale for why AA don't follow polarity and VP trends. The expected effects of solubility on retention (and its relationship with Henry's law constant) should be discussed more broadly, along with lack or solubility information limiting the findings.**

Aqueous solubility is largely dependent on structural properties that this method of untargeted MS is not able to adequately resolve for analysis. The limitation on discussion of solubility stems from the inability to assign high confidence structural information. This is an issue commonly discussed in untargeted complex MS analysis. A comprehensive discussion on this issue would only distract readers from the supported conclusions with speculation that is outside of the current measurement capabilities of this method.

Further, solubility is also already well described in the literature as a piece of the calculation to estimate H*. We were able to avoid some of the issues with calculating H* without high confidence structural assignment by accepting the average difference between the H* for structural isomers as found by Isaacman-Vanwertz and Aumont, 2021 (https://doi.org/10.5194/acp-21-6541-2021). This is discussed in lines 176-180.

**469. 'area'. Typo. I believe you mean surface to 'volume'?**

We have incorporated this correction into the manuscript in line 469.

**490 and 505. 'insulates'.  Wrong word choice. I suggest 'suggests'.**

We have fixed these typos in lines 492 and 505.

References:

Szakáll, M., K. Diehl, S. K. Mitra, and S. Borrmann, 2009: A Wind Tunnel Study on the Shape, Oscillation, and Internal Circulation of Large Raindrops with Sizes between 2.5 and 7.5 mm. J. Atmos. Sci., 66, 755–765, https://doi.org/10.1175/2008JAS2777.1.

Isaacman-Vanwertz, G. and Aumont, B.: Impact of organic molecular structure on the estimation of atmospherically relevant physicochemical parameters, Atmos. Chem. Phys, 21, 6541–6563, https://doi.org/10.5194/acp-21-6541-2021, 2021.

---

## Editor Decision (ED1)

**1. Initial referee #2 comment:** The manuscript presented in the current form is more like a Measurement Report, as its atmospheric implication is not fully discussed or addressed. The only three nighttime samples also limits a broader atmospheric implication. If I understood correctly, the samples were combined before the freezing and determination of retention coefficient. Why not perform the experiments for individual samples, which one can exam the variation of retention for the same compounds?

**Author Response:** There are a variety of reasons why we chose to combine filter samples for our experiment. Using combined samples instead of individual samples eliminates variability to allow for experimental reproducibility and valid statistical evaluation of the retention coefficients. Specifically, some of these confounding variables that are controlled by this method could be large concentration differences between samples, background matrix influences, pH, biasing of trace species, etc. The individual samples on their own also do not produce adequate mass loadings for experimental viability. Combining sample extracts allows for enough liquid volume to be produced for the experiment without unduly diluting the sample. Further, comparative analysis of the individual filter samples is outside objectives of this publication and would also require examining the synoptic conditions during filter sample collection.

**Referee #2 Comment:** The response doesn't answer my questions. My concerns remain unsolved. First, as the authors stated: "Specifically, some of these confounding variables that are controlled by this method could be large concentration differences between samples, background matrix influences, pH, biasing of trace species, etc.". Does this mean that the conclusion from this study may not be applied to aerosol particles from different regions where background matrix can be different from Beijing? Second, If I understand it correctly, the Research articles must include substantial advances and general implications for the scientific understanding of atmospheric chemistry and physics. The authors response: "Further, comparative analysis of the individual filter samples is outside objectives of this publication and would also require examining the synoptic conditions during filter sample collection.". This is also not a good justification. I strongly suggest the authors either present it as a Measurement Report or at least incorporate these above reasons in their response in the manuscript.

**Editor comment:** I agree with the referee.
I acknowledge that you do explain nicely potential atmospheric implications based on your observed trends in the conclusion section. However, the potential caveats and limitations of your study due to the unavoidable mixing of samples should be also discussed. Please address it in the Results and Discussion Section and also in the Conclusions.

**2. Initial Referee #2 Comment:** In the Abstract, the statement in Line 20-22 is overstated as the S- and N-containing compounds are not solely from the NOx and Sox chemistry. The statement in Line 24-26 is not supported by the results of non-sigmoidal relationship or the related discussion. The lower surface-to-volume ratio of the large drop size investigated may be one of the reasons. The differences in the experimental techniques, as I understand, may be the main reason, the wind tunnel and levitation, which have totally different air dynamic environments and surrounding corresponding gas-phase organic concentrations that will affect the gas-liquid diffusion and partitioning, and thus the retention.

**Author Response:** We have amended lines 20-22 to more accurately state what compounds are being referenced. We have amended lines 24-26 to more precisely state the confidence of correlation. We agree that the differing physical parameters of this experiment is a significant barrier to its comparison with wind tunnel studies; we present discussion on this at the end of section 3.3 and in the conclusions

in lines 469-474. One of the objectives of this study is to establish untargeted complex mixture analysis as a method to measure retention before we make more direct comparisons to the wind tunnel. We have future experiments planned as a separate publication involving wind tunnel experiments. With interest for this discussion, our preliminary data from the tunnel shows similar results to this experiment in the levitator. This gives us reason to believe that the physical differences of these experiments is not the cause for the absence of sigmoidal behavior here.

**Referee #2 Comment:** As the authors agree the significant barrier to its comparison with wind tunnel studies, then it is not fair to state that the sigmoidal behavior is an overfitting. In the revision, the authors didn't make any adjustment or include the reasoning in the related discussion. If the authors plan to present in a separate publication involving wind tunnel experiments. I would suggest remove the discussion in the comparison with data from the previous wind tunnel studies.

**Editor comment:** I am still confused how you conclude that N- and S-containing compounds are mostly anthropogenic SOA. There is a huge body of literature that discuss the formation of such compounds also from biogenic sources (isoprene, terpenes etc). If you are sure that your samples contain organics with N and/or S as heteroatoms, just state it like that in the abstract, e.g. "The findings here also indicate that N- and S- containing organics have enhanced retention."
I disagree with the referee to remove the wind tunnel data from Figure 10. However, I would like to see a better explanation for the disagreement of low H* species. Why would different physical parameters lead to bad agreement for low H* species but good agreement for high H* species? Were there any other differences in terms of species properties, e.g. water-solubility (which does not necessarily correlate with H*), polarity, functional groups, molecular weight – just to name a few.

The term 'overfit' has a negative connotation, as obviously also expressed by the referee who calls it 'unfair'. Fact is that low retention of low H* species were measured in the previous study. The red line in Figure 10 seems a good fit to this data. However, whether this data – and therefore the fit - is relevant for atmospheric conditions is a different question. I suggest toning down the reference to 'overfitting' in your text and phrase it more cautiously.

**Comments by Referee #3**
**General editor comment:** Although it is clear that your paper has a companion paper with related experiments, each paper should stand on its own. It cannot be expected that the referee or any reader has to read both papers to understand fundamental findings in the present study. Therefore, please address some of the comments by referee #3 in more detail as suggested below.

**Initial referee #3 comment:** 444-459. The discussion of the relationship with effective Henry's law constant does not adequately address the potential influence of the conditions of freezing and freezing kinetics that are expected to have important impacts on retention coefficients for species with lower effective Henry's constant, as discussed in previous literature. Although the formation of an ice shell as inhibiting expulsion is mentioned at the end here, it appears to be largely an afterthought. The data are consistent with freezing conditions that enhance trapping, increasing retention even for the lowest H* compounds (which are expected to have high variability in retention dependent on freezing conditions). The assumption that a sigmoidal shape is expected, irrespective of freezing conditions does not do the previous literature on the retention phenomena adequate justice. (See for example Stuart and Jacobson 2003, 2004 (cited in this manuscript) and 2005, doi: 10.1007/s10874-006-0948-0).

**Author response:** We have discussed the influence of physical parameters on freezing retention more thoroughly in the Part I publication. https://doi.org/10.5194/egusphere-2024-3917 We agree that the differing physical parameters of this experiment is a significant barrier to its comparison with wind tunnel studies; we present discussion on this at the end of section 3.3 and in the conclusions in lines 469-474. One of the objectives of this study is to establish untargeted complex mixture analysis as a method to measure retention before we make more direct comparisons to the wind tunnel. We have future experiments planned as a separate publication involving wind tunnel experiments. With interest for this discussion, our preliminary data from the tunnel shows similar results to this experiment in the levitator. This gives us reason to believe that the physical differences of these experiments is not the cause for the absence of sigmoidal behavior here.

**Editor comment:** Given that you highlight this large discrepancy between your current study and those that found a sigmoidal R-H* relationship as one of the main findings, it would seem reasonable to include more discussion on previous literature data such as Stuart and Jacobson 2005 and possibly others.

**Initial referee #3 comment:** 453-455 (and 469-473). Although this is an appropriate limitation to discuss, it is too narrow. Why focus only on the potential effect of the surface to volume ratio rather than other conditions of freezing, when there are other factors that have been suggested previously as likely important based on theory and modeling?

**Author response:** We have discussed the influence of physical parameters on freezing retention more thoroughly in the Part I publication. https://doi.org/10.5194/egusphere-2024-3917

**Editor comment:** A brief discussion should be added also in this paper. It does not have to be as detailed in Part I.

**Initial referee #3 comment:** 482- 490. Solubility is expected to influence retention but was not studied here and is only discussed as a rationale for why AA don't follow polarity and VP trends. The expected effects of solubility on retention (and its relationship with Henry's law constant) should be discussed more broadly, along with lack or solubility information limiting the findings.

**Author response:** Aqueous solubility is largely dependent on structural properties that this method of untargeted MS is not able to adequately resolve for analysis. The limitation on discussion of solubility stems from the inability to assign high confidence structural information. This is an issue commonly discussed in untargeted complex MS analysis. A comprehensive discussion on this issue would only distract readers from the supported conclusions with speculation that is outside of the current measurement capabilities of this method.   Further, solubility is also already well described in the literature as a piece of the calculation to estimate H*. We were able to avoid some of the issues with calculating H* without high confidence structural assignment by accepting the average difference between the H* for structural isomers as found by Isaacman-Vanwertz and Aumont, 2021 (https://doi.org/10.5194/acp-21-6541-2021). This is discussed in lines 176-180.

**Editor comment:** I appreciate that you explain here that you explain here why solubility is not taken into account as a parameter. However, this information should be also mentioned in the manuscript text in a couple of sentences. I don't think this would be distractive.
Water solubility and Henry's law constants do not necessarily correlate. The former describes how much of a substance can be dissolved in water and above which limit it forms a solid or supersaturated

solution ; the latter describes how a compound partitions between the gas and aqueous phases. Thus it describes two different thermodynamic concepts.

There are very distinct differences. For example, small alcohols are fully miscible in water but have very low H*; in contrary, some salts are highly water soluble, others not – but all of them basically have infinite Henry's law constants.

=================================================================================

**Additional editor comments**

(line numbers refer to your manuscript version without track-change)

l. 13/14: The structure of this sentence does not seem right; 'incurs' might not be the right verb here. Better use 'causes' or similar.

l. 15: The number and variety of compounds that can form new particles is very limited. Why are so specific here? Do you have evidence that indeed new particle formation occurs in the upper troposphere by compounds released from freezing droplets?

Just the fact that compounds can be released upon vertical transport seems a sufficient motivation to study retention coefficient. These compounds may then take part in any chemical reactions.

l. 25: differs from… (not 'with')

l. 26: I am missing in the abstract a concluding sentence that summarizes why the current study is more reliable/refined/better/… than the previous literature studies.

l. 34: SOA formation is not necessarily part of new particle formation. Compounds with sufficiently low vapor pressure may condense on any particle, whether primary or secondary. Also the initial particle may be inorganic (e.g. by new particle formation due to sulfuric acid) or organic.

l. 49 – 51: I am not convinced that uncertainties in gas transport are the main factor of uncertainty of NPF in the upper troposphere. Isn't it much more the uncertainties in NPF precursors, i.e. which compounds nucleate new particles at what rate?

l. 56: 'incurred' seems wrong here. Do you mean 'occurred'?

l. 74: What is the difference between 'liquid water content' and 'droplet size' here? Isn't the liquid water content simply the water volume and thus proportional to size?

Unless you mean something different, I suggest removing liquid water content here.

l. 82: 'cloud size range' seems odd here. Do you mean 'droplet size range'?

l. 251: Do nitrophenols from other sources (e.g. industrial emissions) have different H/C ratios?

l. 253/254: Are the compounds specific to automobile and shipping emission or could it be also other organosulfates?

l. 256: Sulfonic acid (oxidation state of S = +4) is not sulfate (oxidation state +6)

l. 257: The study by Iinuma et al was conducted at a forested, not urban, site. Given that terpenes are biogenic compounds, I do not think that camphorsulfonic acid is only formed under urban conditions.

l. 266/7 and following: Just providing the sum formula of the compounds is rather confusing since these formula could also represent compounds with functional groups other than amines (cf e.g., https://en.wikipedia.org/wiki/C9H11NO2#) Please either write the formulae such that the molecular structure is clear and/or add the compound names.

l. 277: One of the main oxidation products of isoprene include methyl vinyl ketone (among others), with an H/C ratio of 1.5. Thus, I don't think that the H/C range is a valid indicator of biogenic vs anthropogenic. If I am wrong, please add more recent references demonstrating it.

Tables 2 and 3: These tables could be moved to the supplement as they are not discussed in the text, but oly referred to in a very brief sentence (cf also comment by Referee 3). Their main information is included in Figure 3 . Readers who are interested in the data can find them in supplement.

l. 333/34: CHNOS represents more of the heaviest species in the sample set while CHN is entirely AA. This sentence is quite cryptic. Please reword.

l. 345: The paper by Daito et al does not refer to the atmosphere. The experiments were performed under very different conditions, not relevant to atmospheric ones. Unless you can add a relevant reference here that demonstrates that aqueous phase reactions in the atmosphere lead to NOx removal, I suggest removing this part of the sentence.

l. 419: The paper by Weschler and Nazaroff is about semivolatile organics only and in indoor environments. During the last decade there have been many papers on the volatility ranges of SOA species in the atmosphere. It would seem more appropriate to cite one of them.

Conclusions: Define AA and VP here once more for the readers who only read the conclusion section.

l. 493: 'nitrogen and sulfur inclusion' is not a commonly used term. I think what you mean is 'functional groups containing sulfur or nitrogen'

l. 494/5: cf my comment on the abstract

l. 497: cf my comment about this reference above

l. 501 – 503: Atmospheric chemical processing generally tends to oxidatively degrade large nonpolar species into more water soluble, less volatile species (Iavorivska et al., 2016)

This sentence is not quite correct. Chemical processing does not necessarily lead to degradation of organics but can also lead to functionalization. For this you could cite any atmospheric chemistry textbook rather than the paper by Iavorivska et al that is about deposition.

l. 508/9: Why do you limit your discussion here to SOA precursors? By far not all organics form SOA. Just the fact that less oxidized (or 'fresher') organics are less likely to be retained and vertically transported than more aged organics is an interesting result. Whether they eventually form SOA is not relevant.

l. 515 – 520: I got confused by this text (and obviously also referee 2). What do you want to say here? Would you expect that during summer the WSOC composition is completely different and therefore your results are irrelevant? What is known about WSOC differences between summer and winter?

l. 531ff: Can you explain better what you mean here? What should be included in a transport model? Usually retention coefficients are included for individual species or species groups, independent of their mass distribution. Retention is usually just described as a mass fraction that remains in the ice phase. Why should this depend on species abundancy?
Are you saying that the retention coefficients measured in your study are only valid for this particular composition? If so, why? It would imply that the presence of all compounds in a sample affects the retention of an individual compound.

---

## Author Response (AR3)

We would like to thank the Editor and R#2 for their thorough and productive comments. We would especially like to thank the Editor for their guidance in this discussion and their advocacy on behalf of R#3's comments. We have addressed the Editor and R#2's comments and suggestions in our responses below. Our revised manuscript is uploaded along with a tracked changes document for the handling editor's consideration. For the responses, the lines, figures, and sections referenced match the numbering in the version of the manuscript most uploaded along with this response without tracked changes. The specific references to the manuscript in our response have been highlighted in yellow for convenience.

**Comments made by Referee #2**

**1. Referee #2 Comment: The response doesn't answer my questions. My concerns remain unsolved.**

**First, as the authors stated: "Specifically, some of these confounding variables that are controlled by this method could be large concentration differences between samples, background matrix influences, pH, biasing of trace species, etc.". Does this mean that the conclusion from this study may not be applied to aerosol particles from different regions where background matrix can be different from Beijing?**

**Second, If I understand it correctly, the Research articles must include substantial advances and general implications for the scientific understanding of atmospheric chemistry and physics. The authors response: "Further, comparative analysis of the individual filter samples is outside objectives of this publication and would also require examining the synoptic conditions during filter sample collection.". This is also not a good justification. I strongly suggest the authors either present it as a Measurement Report or at least incorporate these above reasons in their response in the manuscript.**

**Editor comment: I agree with the referee.**

**I acknowledge that you do explain nicely potential atmospheric implications based on your observed trends in the conclusion section. However, the potential caveats and limitations of your study due to the unavoidable mixing of samples should be also discussed. Please address it in the Results and Discussion Section and also in the Conclusions.**

As requested, we have included more discussion on the purpose and limitations of mixing filter extracts in the experiment. The reviewer asks "Does this mean that the conclusion

from this study may not be applied to aerosol particles from different regions where background matrix can be different from Beijing?" They raise a valid concern as there is not enough evidence to assume activity differences due to composition are either negligible or the same elsewhere as in this experiment, but it wouldn't be unreasonable to assume. Afterall, average rainwater DOC is on the order of μM which could be assumed to be dilute enough for activity differences to be negligible. We would also like to distinguish that the concern the reviewer raises is not the result of extract sample mixing, but from unquantified activity differences from sample composition. We have added the following to the Methods (2.1), Results and Discussion section (3.3) and in the Conclusions (4):

> Multiple filters were used to produce the filter extract sample in order to ensure adequate concentration and uniform background signal across multiple measurements. This also controls for potential differences in matrix effects from different filter compositions. 101-103

> Additionally, there are chemical dissimilarities in these experiments, as the present experiment is potentially influenced by activity differences from the extract solution's complexity as opposed to the single or few component solutions used by the previously stated studies. 460-463

> The use of UHPLC-HRMS has allowed for the study of ambient WSOC retention rather than single component or limited mixture experiments from previous studies. While the influence on retention due to activity differences resulting from matrix effects and solution complexity is still unknown, the experiment in this paper demonstrates the viability of UHPLC-HRMS analysis for ambient WSOC and shows the need for further complex mixture study regarding retention. Future studies on retention within hydrometeors should include complex mixture analysis under the physical conditions most similar to the atmosphere, i.e. wind tunnel experiments, smaller droplets, increased ventilation. As this experiment is a first demonstration of retention within a complex mixture, the applicability of the conclusions here to other locations or samplings with different aerosol compositions—thereby potentially different matrix effects —could be challenged. For example, black carbon particles suspended in a drop could strongly bind organic compounds, preventing their transition into the gas phase during freezing; certain surfactant species could change the surface accommodation, inhibiting exchange; or different amounts of inorganic ions could change the ionic strength of the aqueous phase, altering chemical potential. There is not enough evidence to assume these matrix effects are negligible or the same elsewhere as in this experiment, but the assumption is not unreasonable. Rainwater tends to show negligible matrix effects

for other properties and analyses (Pang et al., 2017; Sauret-Szczepanski et al., 2006). Average rainwater DOC is on the order of µM which could be assumed to be dilute enough for activity differences to be negligible compared to pure water solutions. However, these are still unsupported assumptions that are required for broad application of these conclusions. 546-560

On the topic of activity differences due to matrix effects and solution non-ideality, we would like to highlight that there is not even enough evidence present to conclusively state whether the retention of single chemical species is different in a complex mixture versus an idealized single component solution, let alone to try to make direct comparisons of different complex mixtures. This is one of the reasons why we find our manuscript valuable to publication. While there is evidence that suggests solution activity differences might be insignificant—specifically, Borchers et. al. 2024 didn't observe any differences in retention between their single component solutions and a mixture of a few nitrophenols; Gautam et. al. 2025 in Part 1 sees little differences between single and binary mixtures—we can't assume that it either is or isn't significant without presenting more evidence. We intend to explore potential activity differences from matrix effects in future publications, possibly in the way the reviewer suggests, but we must first establish that we can measure freezing retention in a complex solution before we can measure riming retention in a complex solution. Our data adds to the body of evidence to potentially answer the question of activity effects and challenges the current understanding of the relationship between H* and freezing retention. Relegating this publication to a Measurement Report because we controlled for variables that would potentially confound the data overlooks the more apparent findings and the reasons why we controlled for the potential differences in background matrix.

**2. Referee #2 Comment: As the authors agree the significant barrier to its comparison with wind tunnel studies, then it is not fair to state that the sigmoidal behavior is an overfitting. In the revision, the authors didn't make any adjustment or include the reasoning in the related discussion. If the authors plan to present in a separate publication involving wind tunnel experiments. I would suggest remove the discussion in the comparison with data from the previous wind tunnel studies.**

**Editor comment: I am still confused how you conclude that N- and S-containing compounds are mostly anthropogenic SOA. There is a huge body of literature that**

**discuss the formation of such compounds also from biogenic sources (isoprene, terpenes etc). If you are sure that your samples contain organics with N and/or S as heteroatoms, just state it like that in the abstract, e.g. "The findings here also indicate that N- and S- containing organics have enhanced retention."**

**I disagree with the referee to remove the wind tunnel data from Figure 10. However, I would like to see a better explanation for the disagreement of low H\* species. Why would different physical parameters lead to bad agreement for low H\* species but good agreement for high H\* species? Were there any other differences in terms of species properties, e.g. water-solubility (which does not necessarily correlate with H\*), polarity, functional groups, molecular weight – just to name a few.**

**The term 'overfit' has a negative connotation, as obviously also expressed by the referee who calls it 'unfair'. Fact is that low retention of low H\* species were measured in the previous study. The red line in Figure 10 seems a good fit to this data. However, whether this data – and therefore the fit - is relevant for atmospheric conditions is a different question. I suggest toning down the reference to 'overfitting' in your text and phrase it more cautiously.**

The N- and S- containing species that we are trying to reference are the ones we discuss in 3.1 (lines 250-290), specifically the nitro and sulphate species evident via MS2 (Fragments with m/z 62 for nitrate or m/z 80 or 90 for sulfate) that we primarily observe along with the amine-nitrates. The species with low H/C and high DBE are aromatic, indicating they are most likely from fossil fuel/biomass burning. The other nitro and sulphate species tend to be NOx and SO2 products that we believe are from local NOx and SO2 emissions. Since sampling occurred in a very urban environment in the winter, we expect the local NOx and SO2 emissions to be anthropogenically dominant. While the isoprene nitrates that the editor references could be purely biogenic, we find this unlikely under these sampling conditions. To clarify, we are considering the products of biogenic precursors that undergo functionalization with anthropogenically sourced N- or S-, namely from NOx and SO2, to be anthropogenically influenced and thus referred to as an anthropogenic SOA component. For example, the terpene-derived organosulfates that the editor discusses are referenced in lines 263-265. For specificity, we have amended line 21:

> The findings here also indicate that N- and S- containing compounds, primarily nitro and sulfate components of secondary organic aerosols (SOA) anthropogenically related to NOx and SO2 chemistry, have enhanced retention likely due to their increased polarity. 21

Additionally, we have further specified 'SOx' as 'SO2' throughout the manuscript. For the purposes of this publication, the distinction between 'SOx' and 'SO2' is insignificant but the reference of "NOx and SO2 chemistry" is more generally used in literature.

As suggested, we have added more explanation for the disagreement of low H* species with wind tunnel studies and to address the question: "Why would different physical parameters lead to bad agreement for low H* species but good agreement for high H* species?" Along with swapping the term 'overfit' with 'fit' as to not imply an undue negative connotation, we have added this discussion to 3.3:

> Physical differences such as higher surface-to-volume ratio, increased ventilation, or longer freezing times may result in lower retentions that may primarily affect species with lower H* as noted by Jost et al. (2017). Small riming droplets have freezing times in the tens of milliseconds while the drops here take approximately 90 seconds to fully freeze. However, the ice shell formation observed by Gautam et al. 2025 in Part 1 is quite fast, in the range of 5 ms. Specifically, these differences either enhance heat and mass transfer which produces a shorter timescale for expulsion or increases freezing time which allows for a wider range of expulsion timescales. Describing this in the framing of Stuart and Jacobson's (2003, 2004) model, species with larger H* are more likely to be unaffected by these differences as the gas-phase mass transport term and the interfacial mass transport term are still dominated by H* despite the decrease in spread droplet height and increase in thermal velocity such that the total expulsion time is still much longer than the freezing time. 477-484

Additionally, we have added more discussion on the differences in terms of species properties of our dataset and the literature we compare against. This is added to 3.3:

> Further, the species studied in the current literature are mostly inorganics which are very different in terms of solubility, polarity, and molecular size compared to organics studied here, i.e. generally more soluble, more polar, and smaller. This could suggest that the organics measured here should have lower retentions than the inorganics in the literature but that is not observed in the data. However, two of the same nitrophenols studied by Borchers et al. (2024) are measured here: 4-nitrophenol and 2,4-dinitrophenol.. It's also likely that 2-nitrobenzoic acid and 4-nitrocatechol or similar analogues are observed within the dataset since other nitroaromatics that cannot be structurally resolved are observed. 2-nitrophenol is also potentially observed, however 2 and 4 nitrophenol are difficult to distinguish from each other as structural isomers and 2-nitrophenol is not easily ionizable in

this method. However, these assignments cannot be confirmed without better structural information. 463-471

We also edited lines 471-473 to include the omission of formaldehyde from the parametrization by Borchers et al. (2024).

The literature values include only three species with $H^* < 10^4$ and two of them, pinanediol and formaldehyde, are excluded from the parameterization presented by Borchers et al., (2024) as they were perceived as outliers. 471-473

**Comments by Referee #3**

**General editor comment: Although it is clear that your paper has a companion paper with related experiments, each paper should stand on its own. It cannot be expected that the referee or any reader has to read both papers to understand fundamental findings in the present study. Therefore, please address some of the comments by referee #3 in more detail as suggested below.**

**Initial referee #3 comment: 444-459. The discussion of the relationship with effective Henry's law constant does not adequately address the potential influence of the conditions of freezing and freezing kinetics that are expected to have important impacts on retention coefficients for species with lower effective Henry's constant, as discussed in previous literature. Although the formation of an ice shell as inhibiting expulsion is mentioned at the end here, it appears to be largely an afterthought. The data are consistent with freezing conditions that enhance trapping, increasing retention even for the lowest H\* compounds (which are expected to have high variability in retention dependent on freezing conditions). The assumption that a sigmoidal shape is expected, irrespective of freezing conditions does not do the previous literature on the retention phenomena adequate justice. (See for example Stuart and Jacobson 2003, 2004 (cited in this manuscript) and 2005, doi: 10.1007/s10874-006-0948-0).**

**Author response: We have discussed the influence of physical parameters on freezing retention more thoroughly in the Part I publication. https://doi.org/10.5194/egusphere-2024-3917 We agree that the differing physical parameters of this experiment is a significant barrier to its comparison with wind tunnel studies; we present discussion on this at the end of section 3.3 and in the conclusions in lines 469-474. One of the objectives of this study is to establish untargeted complex mixture analysis as a method to measure retention before we make more direct comparisons to the wind tunnel. We have future experiments planned as a separate publication involving wind**

**tunnel experiments. With interest for this discussion, our preliminary data from the tunnel shows similar results to this experiment in the levitator. This gives us reason to believe that the physical differences of these experiments is not the cause for the absence of sigmoidal behavior here.**

**Editor comment: Given that you highlight this large discrepancy between your current study and those that found a sigmoidal R-H\* relationship as one of the main findings, it would seem reasonable to include more discussion on previous literature data such as Stuart and Jacobson 2005 and possibly others.**

We have included more discussion on how the physical factors may account for the nonsigmodial behavior in terms of Stuart and Jacobson (2003, 2004) model. Please consider the previous comment.

**Initial referee #3 comment: 453-455 (and 469-473). Although this is an appropriate limitation to discuss, it is too narrow. Why focus only on the potential effect of the surface to volume ratio rather than other conditions of freezing, when there are other factors that have been suggested previously as likely important based on theory and modeling?**

**Author response: We have discussed the influence of physical parameters on freezing retention more thoroughly in the Part I publication. https://doi.org/10.5194/egusphere-2024-3917**

**Editor comment: A brief discussion should be added also in this paper. It does not have to be as detailed in Part I.**

We have included more discussion on how the physical factors may account for the nonsigmodial behavior in terms of Stuart and Jacobson (2003, 2004) model. Please consider the previous comment.

**Initial referee #3 comment: 482- 490. Solubility is expected to influence retention but was not studied here and is only discussed as a rationale for why AA don't follow polarity and VP trends. The expected effects of solubility on retention (and its relationship with Henry's law constant) should be discussed more broadly, along with lack or solubility information limiting the findings.**

**Author response: Aqueous solubility is largely dependent on structural properties that this method of untargeted MS is not able to adequately resolve for analysis. The limitation on discussion of solubility stems from the inability to assign high confidence structural information. This is an issue commonly discussed in untargeted complex MS analysis. A comprehensive discussion on this issue would only distract readers from the supported conclusions with speculation that is outside of the current measurement capabilities of this method. Further, solubility is also already well described in the literature as a piece of the calculation to estimate H\*. We were able to avoid some of the issues with calculating H\* without high confidence structural assignment by accepting the average difference between the H\* for structural isomers as found by Isaacman-Vanwertz and Aumont, 2021 (https://doi.org/10.5194/acp-21-6541-2021). This is discussed in lines 176-180.**

**Editor comment: I appreciate that you explain here that you explain here why solubility is not taken into account as a parameter. However, this information should be also mentioned in the manuscript text in a couple of sentences. I don't think this would be distractive.**

**Water solubility and Henry's law constants do not necessarily correlate. The former describes how much of a substance can be dissolved in water and above which limit it forms a solid or supersaturated**

**solution ; the latter describes how a compound partitions between the gas and aqueous phases. Thus it describes two different thermodynamic concepts.**

**There are very distinct differences. For example, small alcohols are fully miscible in water but have very low H\*; in contrary, some salts are highly water soluble, others not – but all of them basically have infinite Henry's law constants.**

The limitation and implication of analysis on solubility is already discussed briefly in terms of AA in the above discussed passage. As suggested, we have broadened this discussion with a few more sentences on solubility's influence on H\* and the limitations from the current method:

> Indeed, solubility as a property relevant to retention is applicable to all compounds, specifically in how aqueous solubility is related to H\*. While aqueous solubility is a piece of the typical bond method calculation to estimate H\*, it does not necessarily correlate with true H\*. Further analysis on solubility as a factor for retention would be valuable but the limitations on high confidence structural assignment in this method prevent its thorough investigation. Structure assignment could also

elucidate other retention relevant properties such as hydrogen bonding potential or even acid/base effects. ==515-521==

**l. 13/14: The structure of this sentence does not seem right; 'incurs' might not be the right verb here. Better use 'causes' or similar.**

We have swapped the use of "incurs" to "causes". ==13==

**l. 15: The number and variety of compounds that can form new particles is very limited. Why are so specific here? Do you have evidence that indeed new particle formation occurs in the upper troposphere by compounds released from freezing droplets?**

**Just the fact that compounds can be released upon vertical transport seems a sufficient motivation to study retention coefficient. These compounds may then take part in any chemical reactions.**

This study of retention transport was specifically motivated as an explanation for previously unexplained NPF. Specifically, Williamson et al. (2019) and Bardakov et al. (2021), which we discuss in the introduction in lines ==42-55==, propose the underestimation of available organics as an explanation for NPF in the upper troposphere from convective outflows.

We feel the specific motivation is sufficient, but we have generalized the phrase from "available for new particle formation" to "available for atmospheric processes int the upper troposphere such as new particle formation or ozone formation". ==15==

**l. 25: differs from... (not 'with')**

We have swapped "with" with "from". ==25==

**l. 26: I am missing in the abstract a concluding sentence that summarizes why the current study is more reliable/refined/better/... than the previous literature studies.**

We have added a line at the end of the abstract to summarize the improvements of this study to the literature.

> This study greatly expands on the available experimental measurements of retention by investigating hundreds of compounds in complex chemical conditions more similar to the atmosphere than the previous literature studies. ==27-29==

**l. 34: SOA formation is not necessarily part of new particle formation. Compounds with sufficiently low vapor pressure may condense on any particle, whether primary or secondary. Also the initial particle may be inorganic (e.g. by new particle formation due to sulfuric acid) or organic.**

We have replaced the reference to new particle formation from this line with a reference to inorganic contributions.

> These effects are controlled by factors such as their optical properties, size, and the hygroscopicity (Dusek et al., 2006; Sun et al., 2021), which can change based on the proportions of primary organic aerosols (POA)—directly emitted aerosols—and secondary organic aerosols (SOA)—aerosols formed from the oxidation products of volatile organic compounds (VOC)—as well as inorganic constituents (Hallquist et al., 2009; Liu et al., 2021; Riva et al., 2019). 33-39

**l. 49 – 51: I am not convinced that uncertainties in gas transport are the main factor of uncertainty of NPF in the upper troposphere. Isn't it much more the uncertainties in NPF precursors, i.e. which compounds nucleate new particles at what rate?**

We have edited this line to be more specific to NPF precursors.

> If this NPF is the result of an overlooked mechanism of organic matter transport, it is then critical to elucidate this mechanism for NPF precursors so to constrain uncertainty around the influence of high altitude NPF from convective outflows (Bardakov et al., 2021). 53-55

**l. 56: 'incurred' seems wrong here. Do you mean 'occurred'?**

We believe 'incurred' is used properly here, but we have swapped its use to 'caused'. 60

**l. 74: What is the difference between 'liquid water content' and 'droplet size' here? Isn't the liquid water content simply the water volume and thus proportional to size?**

**Unless you mean something different, I suggest removing liquid water content here.**

The misunderstanding here is that LWC is a property of an individual drop and not of the environment the drop is in. We have struck the phrase "of the droplet" from this line to avoid misunderstanding. (line 78) This phrasing also mistakenly implies ventilation is a property of a single drop.

Liquid water content is more similar to humidity in that it's a mass concentration of water in a volume of air. Droplet size is often related because of growth or evaporation effects.

**l. 82: 'cloud size range' seems odd here. Do you mean 'droplet size range'?**

We mean the size ranges typical for droplets within natural clouds. We have rephrased the line as:

> Additionally, current studies have only examined retention for cloud droplet sizes rather than raindrop sizes. 86

**l. 251: Do nitrophenols from other sources (e.g. industrial emissions) have different H/C ratios?**

I don't believe so but this would be interesting to consider. The H/C ratio is derived from condensed nature of aromatics so all nitrophenols regardless of origin should be similar in terms of H/C ratio.

**l. 253/254: Are the compounds specific to automobile and shipping emission or could it be also other organosulfates?**

The alkylorganosulfates compounds referenced are specific to high sulfur fuel combustion, which is almost exclusive to automobiles and ship emissions. To be specific, saturated organosulfates C>5 are not of biogenic origin, but rather shipping, traffic, or coal combustion. This is discussed in the references presented: Blair et al., 2017; Qi et al., 2021.

If the editor is insinuating the other organosulfates to be biogenic terpene products, we do discuss those in the lines following.

**l. 256: Sulfonic acid (oxidation state of S = +4) is not sulfate (oxidation state +6)**

We have added the phrase "in addition to compounds such as camphorsulfonic acid" as to not mistakenly imply that camphorsulfonic acids is a sulfate. 263

**l. 257: The study by Iinuma et al was conducted at a forested, not urban, site. Given that terpenes are biogenic compounds, I do not think that camphorsulfonic acid is only formed under urban conditions.**

We have amended the line as:

> ...such as camphorsulfonic acid (C10H16O4S, 232 m/z, 9.4 min, L2), which also demonstrate secondary processing with sulfate aerosols (Iinuma et al., 2007; Surratt et al., 2007). 263-264

**l. 266/7 and following: Just providing the sum formula of the compounds is rather confusing since these formula could also represent compounds with functional groups other than amines (cf e.g., https://en.wikipedia.org/wiki/C9H11NO2#) Please**

**either write the formulae such that the molecular structure is clear and/or add the compound names.**

Our method is limited in which structures it can confidently assign. We can assign the sum formula with high confidence and that these compounds are amines based on their ionization mode, H/C ratio, O/C ratio, and any available MS2 spectra. However, a full structure cannot be assigned with high confidence without standards for each compound of interest. We don't want to imply more structural information than we have by giving these compounds more description than the sum formula. We believe that stating the sum formula and class of compound is a more precise description of the present information than presenting a misleading structure candidate. This is consistent with our usage of the convention of Schymanski et al. (2014) to communicate structural confidence.

**l. 277: One of the main oxidation products of isoprene include methyl vinyl ketone (among others), with an H/C ratio of 1.5. Thus, I don't think that the H/C range is a valid indicator of biogenic vs anthropogenic. If I am wrong, please add more recent references demonstrating it.**

The editor is correct in stating that MVK has an H/C ratio of 1.5. However, the distinction of a CHO species with H/C ratio above or below 1.5 is not being used as an indicator of biogenic vs anthropogenic origin in this instance but compound classification of aromatics versus aliphatic acids. In the next line, we are stating that we see few species within a specific range of H/C that the compound classes of humics and lignin that typically have. These classes are then associated with biogenic origin.

We have added a reference to Ma et al. (2022) (Line 287) which more thoroughly investigates the markers of wintertime Beijing aerosol through untargeted MS analysis. This section is primarily intended to describe that our sample set is typical of Beijing winter aerosols.

**Tables 2 and 3: These tables could be moved to the supplement as they are not discussed in the text, but oly referred to in a very brief sentence (cf also comment by Referee 3). Their main information is included in Figure 3 . Readers who are interested in the data can find them in supplement.**

We have moved tables 2 and 3 to the supplement and adjusted line 328 accordingly.

**l. 333/34: CHNOS represents more of the heaviest species in the sample set while CHN is entirely AA.**

**This sentence is quite cryptic. Please reword.**

We have reworded this line for clarity as:

These CHNOS compounds have some of the highest molecular masses in the dataset, evident in Figure 1. The CHN species present are all AA. 340-342

**l. 345: The paper by Daito et al does not refer to the atmosphere. The experiments were performed under very different conditions, not relevant to atmospheric ones. Unless you can add a relevant reference here that demonstrates that aqueous phase reactions in the atmosphere lead to NOx removal, I suggest removing this part of the sentence.**

We have reformulated the sentence with more relevant references:

It is known that NOx participates in the reversible and irreversible uptake of isoprene to aerosol liquid water and can further react with isoprene to produce a substantial number of organonitrates. (El-Sayed et al., 2018; Tsiligiannis et al., 2022). While it's also known that organic nitrogen represents an important fraction of WSOC (Saxena and Hildemann, 1996; Zhang et al., 2002), this data may also indicate that nitrogen chemistry on CHO species enhances their retention in hydrometeors. 351-356

**l. 419: The paper by Weschler and Nazaroff is about semivolatile organics only and in indoor environments. During the last decade there have been many papers on the volatility ranges of SOA species in the atmosphere. It would seem more appropriate to cite one of them.**

We have updated the reference to two more recent publications on LVOCs and IVOCs. (Li et al., 2023; Manavi and Pandis, 2024). 427

**Conclusions: Define AA and VP here once more for the readers who only read the conclusion section.**

We have made this adjustment in lines 497 and 509.

**l. 493: 'nitrogen and sulfur inclusion' is not a commonly used term. I think what you mean is 'functional groups containing sulfur or nitrogen'**

We have incorporated this into the manuscript in line 526.

**l. 494/5: cf my comment on the abstract**

We have adjusted this sentence for clarity and specificity in agreement with the previous.

Overall, the high retentions of the nitro and sulfate species typically anthropogenically related to NOx and SO2 chemistry, indicate that $NO_x$ and $SO_x$

chemistry may enhance the retention of these SOA species, reducing their likelihood of reaching the upper atmosphere. ==527-529==

**l. 497: cf my comment about this reference above**

We have reformulated the sentence with more relevant references as previously done:

> Further on this, other studies have demonstrated that NOx participates in the reversible and irreversible uptake of isoprene to aerosol liquid water and can further react with isoprene to produce a substantial number of organonitrates. (El-Sayed et al., 2018; Tsiligiannis et al., 2022). ==529-532==

**l. 501 – 503: Atmospheric chemical processing generally tends to oxidatively degrade large nonpolar species into more water soluble, less volatile species (Iavorivska et al., 2016)**

**This sentence is not quite correct. Chemical processing does not necessarily lead to degradation of organics but can also lead to functionalization. For this you could cite any atmospheric chemistry textbook rather than the paper by Iavorivska et al that is about deposition.**

We have edited the sentence to include functionalization and updated the reference to "Atmospheric Chemistry and Physics: From Air Pollution to Climate Change" by John H. Seinfeld and Spyros N. Pandis.

> Atmospheric chemical processing generally tends to functionalize or oxidatively degrade large nonpolar species into more water-soluble, less volatile species (John H. Seinfeld and Spyros N. Pandis, 2019). ==537-539==

**l. 508/9: Why do you limit your discussion here to SOA precursors? By far not all organics form SOA. Just the fact that less oxidized (or 'fresher') organics are less likely to be retained and vertically transported than more aged organics is an interesting result. Whether they eventually form SOA is not relevant.**

We have broadened the discussion here by swapping the instances of 'SOA precursors' to simply 'organics'.

> This indicates that many freshly oxidized organics may have a lower potential to be retained than aged organics and may generally suggest that freshly oxidized organics are more likely to reach the upper atmosphere than primary organics or aged organics. ==543-545==

**l. 515 – 520: I got confused by this text (and obviously also referee 2). What do you want to say here? Would you expect that during summer the WSOC composition is**

**completely different and therefore your results are irrelevant? What is known about WSOC differences between summer and winter?**

This bit of discussion was added in response to a concern raised by Reviewer #1. They noted that the conditions required for the retention transport that we describe occur more frequently in the summer rather than in winter, which is when our samples were taken. Specifically, the convective systems that would facilitate the transport of surface emissions to altitudes where ice cloud formation occurs are not typical for the winter. That's not to limit the relevance of our results, but to highlight that our results may not be entirely representative of all the relevant species that participate in this process.

We have expanded on this discussion with the following:

> Additionally, broader application of these conclusions should consider the differences in relevant WSOC composition regarding seasons. This sample set is representative of winter aerosols, which show a high contribution from nitroaromatic compounds as well as a lower degree of oxidation and a lower proportion of organosulfates in comparison to the summer (Ma et al., 2022). 562-564

**l. 531ff: Can you explain better what you mean here? What should be included in a transport model? Usually retention coefficients are included for individual species or species groups, independent of their mass distribution. Retention is usually just described as a mass fraction that remains in the ice phase. Why should this depend on species abundancy?**

**Are you saying that the retention coefficients measured in your study are only valid for this particular composition? If so, why? It would imply that the presence of all compounds in a sample affects the retention of an individual compound.**

The discussion here was aimed at qualifying the application of the parameterizations presented in Table S1 and S2. The parameterizations simply provide the distribution for all measured species without regard to which species are more abundant. It most likely is best practice to implement the retention coefficients as individual species or species groups, but we wanted to make that distinction for someone looking to estimate the retention of a random organic by applying the frequency distribution of this dataset.

To clarify this, we have swapped out the instances of 'measurements' and 'data' with more specific phrases:

> These parameterizations of retention also present the distribution of retention coefficients for the variety of species present and not necessarily the mass

distribution of species potentially present in the atmosphere. Corrections for species abundancy must first be made in order to apply the frequency distribution of the retention coefficients here to organic transport models. 580-584

**References**

Borchers, C., Seymore, J., Gautam, M., Dörholt, K., Müller, Y., Arndt, A., Gömmer, L., Ungeheuer, F., Szakáll, M., Borrmann, S., Theis, A., Vogel, A. L., and Hoffmann, T.: Retention of α-pinene oxidation products and nitro-aromatic compounds during riming, https://doi.org/10.5194/egusphere-2024-1443, 2024.

Li, Z., Hyttinen, N., Vainikka, M., Tikkasalo, O. P., Schobesberger, S., and Yli-Juuti, T.: Saturation vapor pressure characterization of selected low-volatility organic compounds using a residence time chamber, Atmos Chem Phys, 23, 6863–6877, https://doi.org/10.5194/ACP-23-6863-2023, 2023.

Manavi, S. E. I. and Pandis, S. N.: Contribution of intermediate-volatility organic compounds from on-road transport to secondary organic aerosol levels in Europe, Atmos Chem Phys, 24, 891–909, https://doi.org/10.5194/ACP-24-891-2024, 2024.

Pang, X., Lewis, A. C., and Shaw, M. D.: Analysis of biogenic carbonyl compounds in rainwater by stir bar sorptive extraction technique with chemical derivatization and gas chromatography-mass spectrometry, J Sep Sci, 40, 753–766, https://doi.org/10.1002/JSSC.201600561;JOURNAL:JOURNAL:15214168;CTYPE:STRING:JOURNAL, 2017.

Sauret-Szczepanski, N., Mirabel, P., and Wortham, H.: Development of an SPME–GC–MS/MS method for the determination of pesticides in rainwater: Laboratory and field experiments, Environmental Pollution, 139, 133–142, https://doi.org/10.1016/J.ENVPOL.2005.04.024, 2006.